# Field experiment for open-loop yaw-based wake steering at a commercial onshore wind farm in Italy

Bart M. Doekemeijer[1], Stefan Kern[2], Sivateja Maturu[2,4], Stoyan Kanev[3], Bastian Salbert[4], Johannes Schreiber[4], Filippo Campagnolo[4], Carlo L. Bottasso[4], Simone Schuler[2], Friedrich Wilts[5], Thomas Neumann[5], Giancarlo Potenza[6], Fabio Calabretta[6], Federico Fioretti[6], and Jan-Willem van Wingerden[1]

[1]Delft Center for Systems and Control, Delft University of Technology, Delft, The Netherlands
[2]GE Renewable Energy, D-85748 Garching bei München, Germany
[3]TNO, Energy Transition, Westerduinweg 3, 1755 LE, Petten, The Netherlands
[4]Wind Energy Institute, Technische Universität München, 85748 Garching, Germany
[5]UL International GmbH - DEWI, Ebertstrasse 96, Wilhelmshaven, Germany
[6]ENEL Green Power, Viale Regina Margherita 125, Rome, Italy

**Correspondence:** Bart Doekemeijer (b.m.doekemeijer@tudelft.nl)

**Abstract.** The concept of wake steering in wind farms for power maximization has gained significant popularity over the last decade. Recent field trials described in the literature demonstrate the real potential of wake steering on commercial wind farms, but also show that wake steering does not yet consistently lead to an increase in energy production for all inflow conditions. Moreover, a recent survey among experts shows that validation of the concept remains the largest barrier for adoption currently.

In response, this article presents the results of a field experiment investigating wake steering in three-turbine arrays at an onshore wind farm in Italy. This experiment was performed as part of the European CL-Windcon project. While important, this experiment excludes an analysis of the structural loads and focuses solely on the effects of wake steering on power production. The measurements show increases in power production of up to $35\%$ for two-turbine interactions and up to $16\%$ for three-turbine-interactions. However, losses in power production are seen for various regions of wind directions too. In addition to the

gains achieved through wake steering at downstream turbines, more interesting to note is that a significant share in gains are from the upstream turbines, showing an increased power production of the yawed turbine itself compared to baseline operation for some wind directions. Furthermore, the surrogate model, while capturing the general trends of wake interaction, lacks the details necessary to accurately represent the measurements. This article supports the notion that further research is necessary, notably on the topics of wind farm modeling and experiment design, before wake steering will lead to consistent energy gains

in commercial wind farms.

## 1   Introduction

Over the last years, the concept of wake steering in wind farms has gained significant popularity in the literature (Boersma et al., 2017; Kheirabadi and Nagamune, 2019). Fundamentally, wake steering leverages the principle that intentional yaw misalignment of a wind turbine displaces its downstream wake. Thus, by choosing the right yaw misalignment, the wake

formed by an upstream turbine can be directed away from a downstream turbine at the cost of a small reduction in its own power production and a change in mechanical loads on the turbine structure. Consequently, this concept enables a net increase in the power production of downstream turbines and, at large, wind farms. In high-fidelity simulations, wake steering strategies are shown to increase the wind-farm-wide power production by $15\%$ for wake-loss-heavy situations (e.g., Gebraad et al., 2016). Moreover, wind tunnel experiments indicate increases in the wind farm's power production of up to $4-12\%$ for two-turbine arrays (Adaramola and Krogstad, 2011; Schottler et al., 2016; Bartl et al., 2018), up to $15-33\%$ for three-turbine arrays (Campagnolo et al., 2016a, b; Park et al., 2016), and up to $17\%$ for a five-turbine array (Bastankhah and Fernando, 2019). However, these experiments neglect realistic wind variability and measurement uncertainty; often, the wind direction is known a priori and fed directly to the controller. A field experiment of wake steering in a scaled wind farm by Wagenaar et al. (2012) is inconclusive compared to baseline operation. In response, there has been a surge in the interest towards the development of reliable wake steering solutions that address issues of wind variability and measurement uncertainty (e.g., Rott et al., 2018; Simley et al., 2019; Kanev, 2020; Doekemeijer et al., 2020). Additionally, interest towards the effect of yaw misalignment on the turbine structural loads is rising, with publications showing both reductions and increases in structural loads, depending on the turbine component, misalignment angle and wind profile (e.g., Kragh and Hansen, 2014; Damiani et al., 2018; Ennis et al., 2018). The scope of this article is limited to the effects of wake steering on power production.

A small number of articles focus on the validation of wake steering for power maximization at full-scale turbines and commercial wind farms. Fleming et al. (2017a) instrumented a GE 1.5MW turbine with a lidar and operated the turbine at various yaw misalignments to study the wake deflection downstream. Then, Fleming et al. (2017b) demonstrated wake steering at an offshore commercial wind farm with relatively large turbine spacing of 7 to 14 times the rotor diameter ($7$–$14D$). These field trials involved yawing an upstream wind turbine and investigating the change in power production at the downstream turbine. When looking at two turbine pairs spaced $7D$ and $8D$ apart respectively, a gain was seen in the power production of the second turbine for most wind directions, at the cost of a much smaller loss on the upstream machine. This led to an increase in the combined power production of up to $10\%$ for various wind directions. No significant improvements were seen for third turbine pair spaced at $14D$. However, the uncertainty bounds remain fairly large and the results also suggest that the net energy yield reduces due to wake steering for a smaller number of cases. Thereafter, Fleming et al. (2019, 2020) evaluated wake steering at a closely spaced ($3$–$5D$) onshore wind farm surrounded by complex terrain, again considering two-turbine interactions. Measurements show that the net energy yield can increase by up to $7\%$ and reduce by up to the same amount for the $3D$-spaced turbine pair, depending on the wind direction. Similarly, the change in the net energy yield for the $5D$-spaced turbine pair is between $+3\%$ and $-2.5\%$. It must be noted that the situations that lead to an increase in power production outnumber those that show a decrease in power production. Furthermore, Howland et al. (2019) assessed the concept of wake steering on an onshore 6-turbine wind farm with $3.5D$ turbine spacing. While significant gains in power production of up to $47\%$ for low wind speeds and up to $13\%$ for higher wind speeds are reported for particular situations, the authors also state that the net energy gain of the wind farm over annual operation is negligible compared to baseline operation.

The current literature on wake-steering field experiments suggests that wake steering has real potential to increase the net energy production in wind farms, yet does not consistently lead to an increase in power production for all inflow conditions.

Moreover, only Howland et al. (2019) address multiple-turbine interaction, rather than the two-turbine interactions addressed in Fleming et al. (2017b, 2019, 2020). Clearly, additional research and validation is necessary for the industry-wide adoption of wake steering control algorithms for commercial wind farms. This is in agreement with a recent survey among experts in academia and industry working on wind farm control (van Wingerden et al., 2020), which shows that the lack of validation is currently the primary barrier preventing implementation of wind farm control.

In this regard, this article presents the results of a field campaign for wake steering at an onshore wind farm with complex terrain in Italy, as part of the European CL-Windcon project (European Commission, 2020). The goal of this experiment is to assess the potential of the current wake steering strategies for such complicated, commercial wind farms. The contributions of this work are:

- As one of the few in the literature, demonstrating the potential of a state-of-the-art wind farm control algorithm for wake
steering at an commercial onshore wind farm with complex terrain.

- Investigating wake interactions in non-aligned (i.e., not in a straight line) three-turbine arrays, in which yaw misalignments are applied to the first two turbines. The yaw misalignments are computed offline, based on the optimization of a simplified mathematical model of the wind farm. Wake steering for non-aligned turbine arrays has not been treated in the existing field experiments.

- The assigned yaw misalignment covers both negative and positive angles, depending on the wind direction. In the existing literature, turbines were only misaligned in one direction.

- Addressing multiple turbine types. Namely, the second turbine, WTG E5, has a different hub height and rotor diameter than the other turbines. This has not yet been assessed in the existing field experiments.

The article is structured as follows. Section 2 outlines the wind farm and the experiment. Section 3 shows the turbine control
setpoints, calculated using state-of-the-art wind farm control solutions. Section 4 describes the data post-processing. Section 5 presents the results of the field experiment. Finally, the article is concluded in Section 6.

## 2  Methodology

This section outlines the details of the experiment. In Section 2.1, the wind farm layout, terrain, and turbine properties are depicted. Then, Section 2.2 addresses the wake steering experiment itself and discusses several challenges faced compared to
previous field tests. Finally, Section 2.3 describes what data is collected during the experiment.

### 2.1  The wind farm

The wake-steering field campaign has been executed on a subset of turbines in a commercial, onshore wind farm near Sedini on the island of Sardinia, Italy. The field experiment is part of the European CL-Windcon project. The wind farm, owned and operated by ENEL Green Power (EGP), is typically operated for commercial purposes, not for testing. The wind farm contains

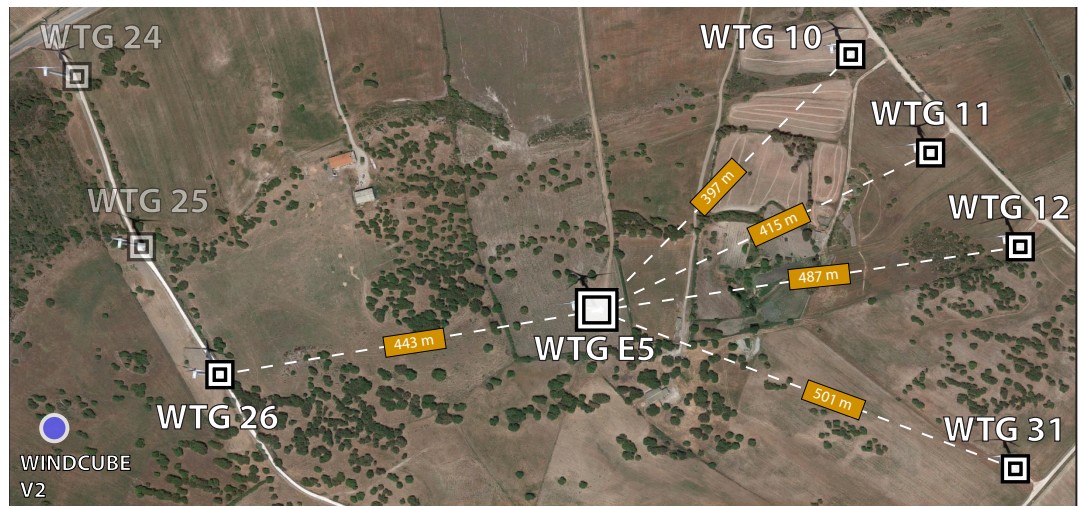

**Figure 1.** Positions of the wind turbines used in the wake steering campaign. Turbines WTG 26 and E5 are operated at a yaw misalignment to steer the wakes away from downstream turbines WTG E5, 10, 11, 12, and 31. WTG 25 is used for normalization. WTG E5 is a GE 1.5sle turbine, and all others are GE 1.5s turbines. A WindCube V2 lidar system is used to characterize the inflow in front of WTG 26 for a short period of the field campaign. Imagery ©2020 Google, Imagery ©2020 CNES / Airbus, Maxar Technologies, Map data ©2020.

**Table 1.** General properties of the GE 1.5s and GE 1.5sle wind turbines

| Variable | GE 1.5s | GE 1.5sle |
|---|---|---|
| Rated power (MW) | 1.5 | 1.5 |
| Cut-in wind speed (m/s) | 4.0 | 3.5 |
| Rated wind speed (m/s) | 13.0 | 12.0 |
| Rotor diameter (m) | 70.5 | 77.0 |
| Hub height (m) | 65 | 80 |

a total of 43 GE wind turbines, of which 36 turbines are of the type GE 1.5s, and 7 turbines of the type GE 1.5sle. Properties of the two turbine types found in this farm are listed in Table 1. The relevant subset of the wind farm layout is shown in Figure 1. In the wake steering campaign, WTG E5 is of the type GE 1.5sle, and all other turbines are of the type GE 1.5s.

The Sedini wind farm is located in a relatively flat area with an average elevation of 360 m to 400 m above sea level, surrounded by hills of $400 - 450$ m above sea level. The site vegetation consists of scrub and clear areas. The predominant

wind directions are from the west and south-east. The mean wind speed is $4 - 6$ m/s, depending on the season. The site has a median ambient turbulence intensity of $15 - 25\%$ with a mean shear exponent of 0.05 to 0.25 for day and night, respectively (Kern et al., 2017). Figure 2 shows the estimated wind direction, wind speed, and turbulence intensity of the data collected by the upstream turbines.

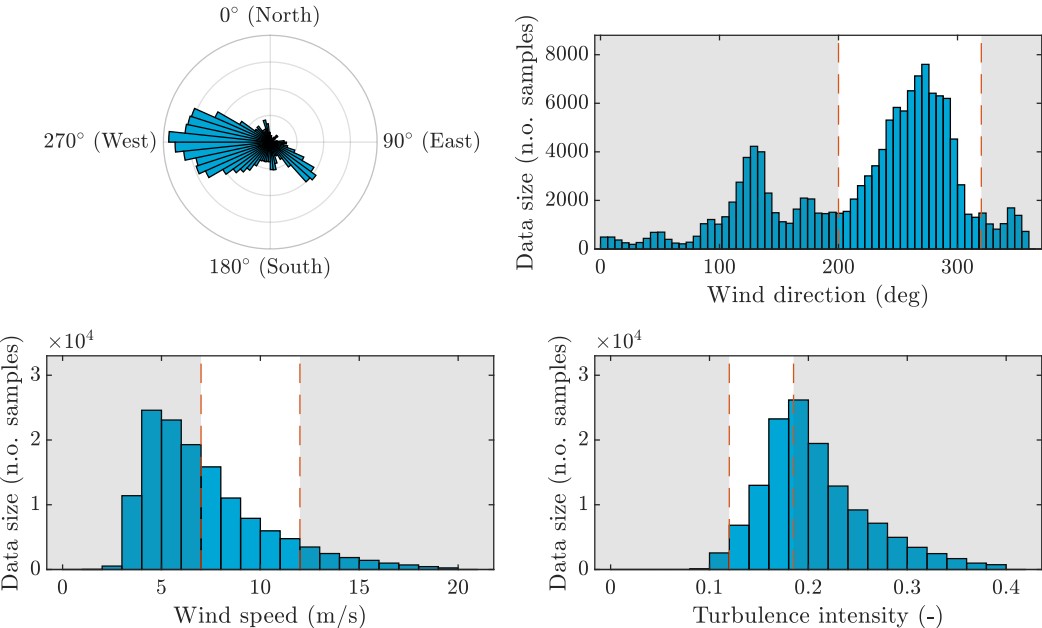

**Figure 2.** All measured data from 19 August 2019 until 3 February 2020, binned by wind direction, wind speed, and turbulence intensity. Wind comes predominantly from the west, which is within the scope of the wake steering experiment. Furthermore, wind speeds are relatively low and turbulence intensities are high. The gray area covers data that is discarded in analysis of the wake steering experiments.

## 2.2 Experiment design

For the wake steering experiments, eight turbines are used: WTG 10, 11, 12, 24, 25, 26, 31, and E5, as shown in Figure 1. The situations of interest are when WTG 26 sheds a wake on WTG E5 and one or both turbines shed wakes on turbines WTG 10, 11 or 12. Additionally, for north-west wind directions, the situation where turbine WTG E5 sheds a wake on WTG 31 is of interest. For all situations, WTG 25 is used as a reference turbine, and WTG 24 and WTG 25 are used to estimate the inflow ambient conditions for WTG 26 and WTG E5. While this layout lends itself well to wake steering, this field campaign faces several challenges, namely:

- Part of the experiment is in late summer, with higher turbulence levels and lower wind speeds compared to winter. Moreover, onshore wind farms typically experience a higher turbulence intensity than offshore farms. Higher turbulence levels generally yield lower benefits for yaw-based wake steering (Appendix A).

- There are variations in the terrain, turbine hub heights, and turbine rotor diameters throughout the wind farm. Specifically, hilly terrain is likely to contribute to variations in the ambient wind speed and wind direction between different upstream wind turbines. However, almost all surrogate models in the literature assume a uniform (homogeneous) ambient inflow, where each upstream turbine experiences the same wind speed, wind direction and turbulence intensity (Boersma et al.,

**Table 2.** Wind turbines of interest, scheduled according to the wind direction. To maximize the benefits of wake steering, only three turbines are considered at a time, depending on the ambient wind direction.

| Wind direction | Turbines of interest |
|---|---|
| $< 235°$ | WTG 26, WTG E5, and WTG 10 |
| $235° - 253°$ | WTG 26, WTG E5, and WTG 11 |
| $253° - 276°$ | WTG 26, WTG E5, and WTG 12 |
| $\geq 276°$ | WTG 26, WTG E5, and WTG 31 |

2017). Variations in the ambient wind direction have a large influence on wakes, and thereby on the wake steering campaign. Additionally, as can be seen in Figure 1, various types of vegetation are present on the ground. The surface roughness varies with the type of vegetation, which in turn impacts the level of turbulence and thereby wake recovery. Due to its high level of complexity, surrogate models address these effects to a very limited degree and lack validation with higher-fidelity and experimental data. The surrogate model used in this work will be discussed in Section 3.2.

– The downstream turbines are closely spaced, implying that gains due to wake steering are hardly noticeable when considering the complete downstream array. For example, if the wake of WTG E5 is redirected away from WTG 10, then the combined net gain of WTG 26, E5, 10, 11, 12 and 31 would be relatively small. In addition, wake steering should be very precise, as the wake must be redirected in between WTG 10, 11, 12, and 31 to lead to a net energy increase. For example, if the wake is deflected away from WTG 11, it may be moved on top of WTG 10 or 12, thereby effectively leading to zero net gain.

– The ambient conditions are to be estimated using existing turbine sensors, rather than external measurement equipment such as a lidar system. This is likely to be less accurate but more realistic for the future commercialization of wake steering.

These challenges, in addition to common challenges such as irregular turbine behavior and measurement uncertainty, have led to the decision to consider only one of the downstream turbines (WTGs 10, 11, 12, 31) at a time, scheduled according to the ambient wind direction, as listed in Table 2. Thus, the remaining downstream turbines are ignored in the analysis. This means that the wake can be steered away from the considered turbines and onto the ignored turbines. This is exemplified in Figure 3, depicting what wake interactions are considered per wind direction.

## 2.3 Data acquisition

The benefit of wake steering strongly depends on the ambient conditions. Therefore, it is important to accurately characterize these inflow conditions. In this field campaign, data is acquired from a number of sources. A met mast with a height of $63.5$ m is installed $0.5$ km north of WTG 25. The met mast provides information about the wind speed, wind direction, vertical shear, temperature, and humidity in the wind farm. However, ambient conditions vary significantly throughout the farm, not in the

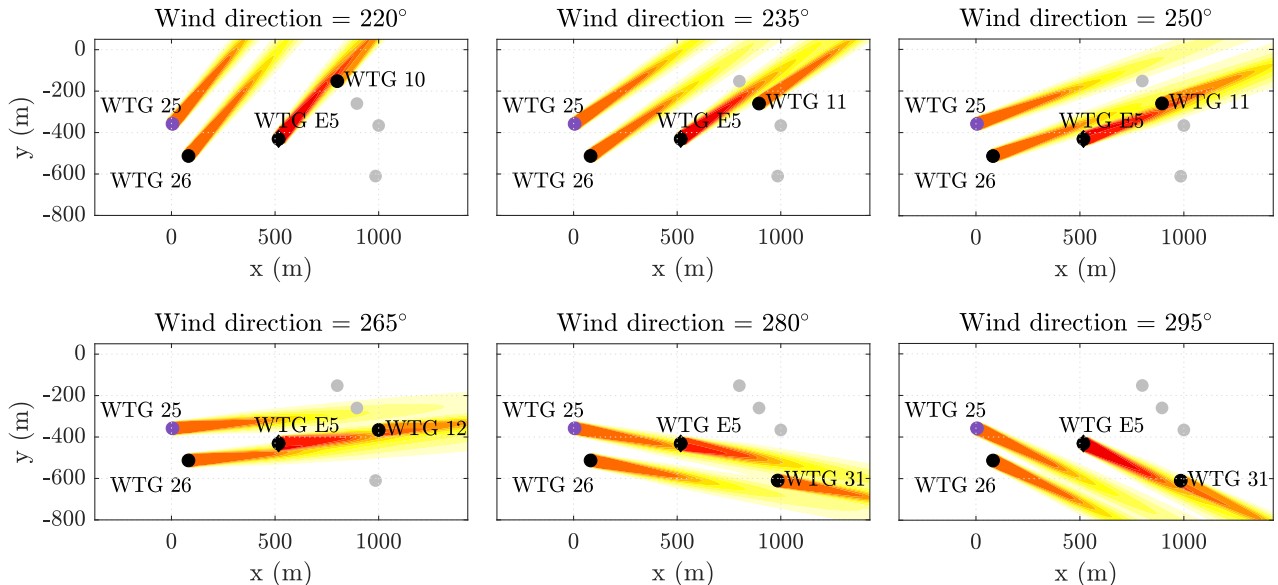

**Figure 3.** Predicted flow fields for various wind directions in baseline operation. To maximize the benefits of wake steering, only three turbines are considered at a time, depending on the ambient wind direction. The considered turbines are WTG 26, WTG E5, and one of the downstream turbines (operated without yaw misalignment). The schedule of which turbines are considered is listed in Table 2. Note that this figure is shown for explanatory purposes and therefore the simulation setup is not described in detail.

least due to this being an onshore wind farm. For this reason, a mobile, ground-based vertical lidar system of the type Leosphere WindCube V2 is installed to measure the inflow at WTG 26 for the first several months of the wake steering field campaign, as shown in Figure 1. The WindCube is installed at an estimated distance of $3D$ in front of WTG 26, thereby lying outside of the turbine's induction zone. This lidar system measures the wind speed at a 0.1 m/s accuracy and the wind direction with a 2° accuracy at 12 programmable heights up to 200 m, with a sampling rate of 1 Hz. This lidar system cannot communicate with the control algorithm in real time and thus was only used in postprocessing to validate the ambient wind conditions estimated in front of WTG 26 using WTG 24 and WTG 25. The validation is shown in Figure 4, displaying a good fit for the wind speed. Note that a bias is seen in the wind direction estimates.

In addition to the lidar system, WTG 26 and WTG E5 are instrumented with an additional, accurate nacelle anemometer. Also, WTG 12, 26, and E5 are each instrumented with an additional, accurate nacelle position sensor. Note that these sensors were only available during the first months of the field experiment, used for calibration and monitoring. The GE wind turbines provide standardized SCADA data such as the generator power, the wind speed measured by the anemometer, the wind direction measured by the wind vane, and the yaw orientation measured with the yaw sensor. An algorithm internal to the GE turbine provides estimates of the 1-minute-averaged wind speed, 1-minute-averaged wind direction, and 10-minute-averaged turbulence intensity.

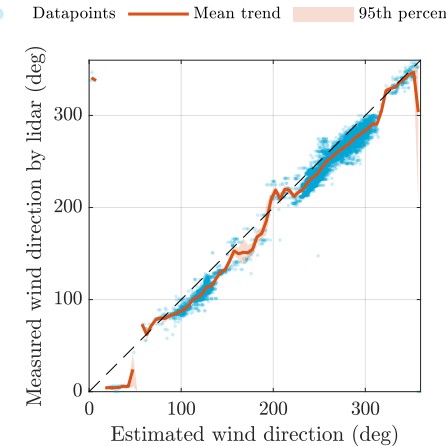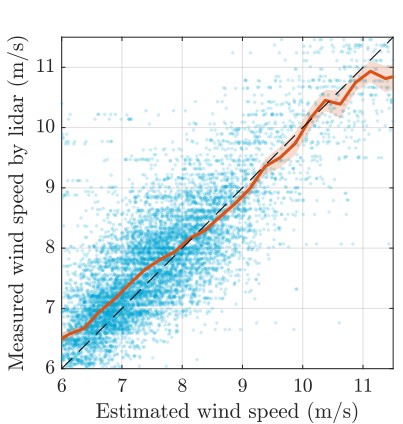

**Figure 4.** Comparison of wind direction and wind speed estimates from the lidar (10-minute averages) and from the turbine anemometers (1-minute averages). For the field campaign, the freestream wind direction, wind speed and turbulence intensity at WTG 26 are estimated using upstream turbines WTG 25 and WTG 24. This approach is validated by comparing the estimates to measurements of the Leosphere WindCube V2 lidar, installed in front of WTG 26 throughout the first several months of the field campaign. The figure shows that the estimates largely match the measurements and the 95% uncertainty bounds, denoted by the shaded region, are narrow. Note that waked sectors (e.g., zone at $80 - 90°$) are not removed in these plots.

## 3 Controller synthesis

As the research field in wind farm control is quickly evolving, an increasing amount of focus is put on closed-loop wind farm control solutions (Doekemeijer et al., 2019). However, implementing and testing such a closed-loop wind farm control algorithm is not feasible for the designated field campaign and instead an open-loop solution is opted for. Closed-loop solutions require additional communication infrastructure compared to open-loop solutions. Also, the actual turbine behavior becomes less predictable as the complexity of the controller increases significantly.

The controller consists of two components. Firstly, the ambient conditions (being the wind direction, wind speed, turbulence intensity) are estimated. How these variables are estimated is described in Section 3.1. Secondly, the optimal turbine yaw setpoints for WTG 26 and WTG E5 are interpolated from a three-dimensional look-up table using the estimated atmospheric conditions. The synthesis of this three-dimensional look-up table is outlined in Section 3.2.

### 3.1 Estimation of the ambient conditions

As outlined in Section 2.3, the ground-based lidar cannot be used in real-time for the wind farm control solution. Moreover, the met mast is located too far away to give a reliable estimate of the ambient conditions. Therefore, turbine SCADA data is used to derive an averaged freestream wind speed, wind direction, and turbulence intensity for the inflow of WTG 26. This estimated wind direction is also assumed to be the wind direction at WTG E5. To obtain the ambient wind condition estimate

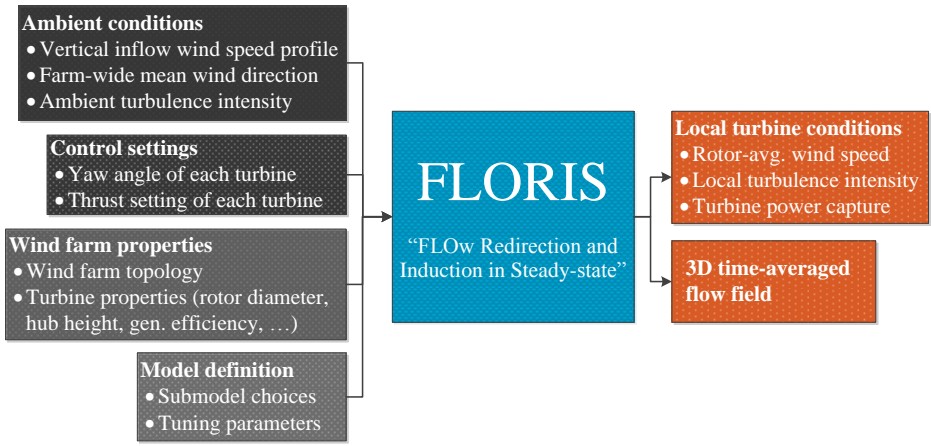

**Figure 5.** Flowchart of the FLORIS model. This model has four classes of inputs: the ambient conditions, a set of model parameters, the turbine control settings, and the wind farm properties (e.g., layout). FLORIS maps these inputs in a static fashion to a set of turbine outputs being the power capture and the three-dimensional flow field.

in front of WTG 26, the individual estimates from turbines WTG 24 and WTG 25 are averaged, which operate in freestream flow for the wind direction range considered for the wake steering experiments. Note that a bias in the wind direction estimate was previously seen in Figure 4. Rather than using the lidar which is likely prone to bias and uncertainty, this is corrected for by comparing the estimated position (wind direction) of the largest power deficits at downstream turbines from FLORIS to the SCADA measurements.

### 3.2 Optimization of the turbine control setpoints

The turbine yaw angles are optimized using the FLOw Redirection and Induction in Steady-state (FLORIS) surrogate model, developed by CU Boulder, NREL and the Delft University of Technology (Gebraad et al., 2016; Doekemeijer and Storm, 2019). FLORIS is a surrogate wind farm model that combines several submodels from the literature, such as the single-wake model from Bastankhah and Porté-Agel (2016), the turbine-induced turbulence model by Crespo and Hernández (1996), and the wake superposition model by Katic et al. (1987). The surrogate model predicts the steady three-dimensional flow field and turbines' operating conditions of a wind farm under a predefined inflow at a low computational cost in the order of 10 ms to 1 s. Note that FLORIS has been fit to high-fidelity simulation data previously (Doekemeijer et al., 2019), and therefore inherently includes the time-averaged effects of dynamic flow behavior such as wake meandering. Figure 5 shows a flowchart of the inputs and outputs of FLORIS.

The yaw angles of WTG 26 and E5 were optimized in FLORIS for a range of wind directions ($200°$ to $320°$ in steps of $2°$), wind speeds ($3$ m/s to $13$ m/s in steps of $1$ m/s), and turbulence intensities ($7.5\%$, $13.5\%$, and $18.0\%$). Note that the optimization was done using the wind-direction-scheduled layout as described in Section 2.2. The optimization took approximately $10^2$ CPU hours. The yaw angles are were then averaged and fixed between wind speeds of $5$ m/s and $11$ m/s in postprocessing to reduce

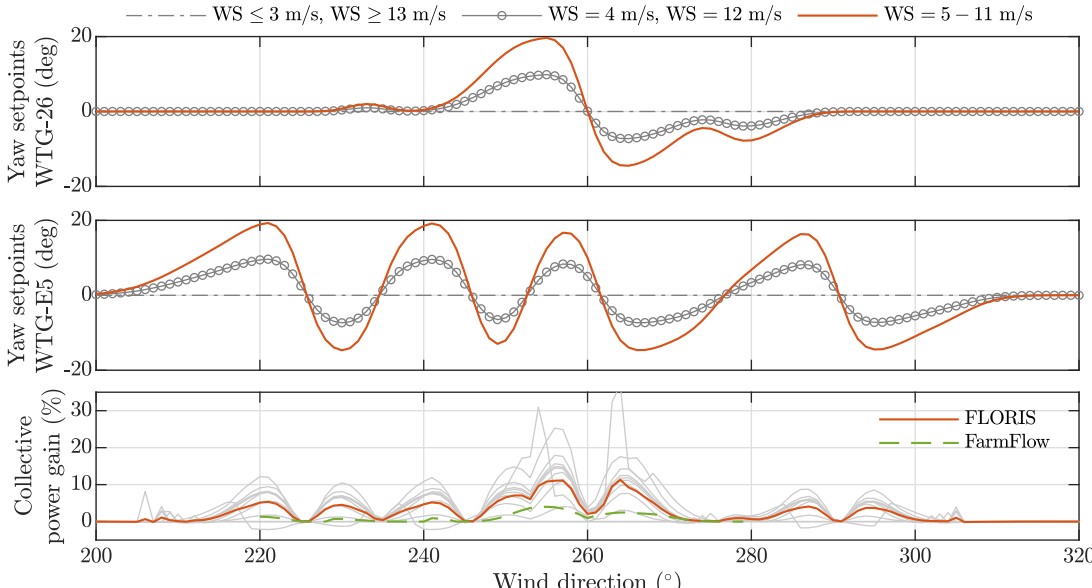

**Figure 6.** The turbine yaw setpoints for WTG 26 and WTG E5 for a freestream turbulence intensity of 7.5%. The yaw angles hold constant values for wind speeds of 5 m/s to 11 m/s. At lower respectively higher wind speeds, the setpoints are interpolated to a yaw angle of $\gamma = 0°$ at 3 m/s and 13 m/s. The collective power gain of WTG 26, WTG E5, and the downstream turbine (WTG 10, 11, 12, or 31) averaged over all wind speeds is shown as the solid orange line (FLORIS) and the green dashed line (FarmFlow) in the bottom plot. The gray lines therein represent the predicted gains for one wind speed by FLORIS.

yaw actuation at a negligible loss in the expected gains, verified by simulations in FLORIS and supported by findings from the literature (Kanev, 2020). Below wind speeds of 5 m/s and above 11 m/s, the angles are interpolated linearly to a yaw angle of $\gamma = 0°$ at 3 m/s and 13 m/s, respectively. This is to avoid undesirable behavior near cut-in and rated operation.

Furthermore, to reduce sensitivity of the optimized yaw setpoints to the wind direction, a Gaussian smoothing kernel was applied to the table of optimized setpoints with a standard deviation of $1.5°$. This is necessary because, when sweeping over the wind direction, there are situations in which it would be better to displace a wake to the other side of a downstream turbine. This results in a discontinuous change in the yaw misalignment (Rott et al., 2018). A better solution would be hysteresis (e.g., Kanev, 2020), but this is not possible in the current framework of the turbine manufacturer. The smoothed look-up table for a turbulence intensity of 7.5% is shown in Figure 6. This figure also shows the predicted gains in power capture for the specified subset of turbines according to FLORIS in idealized conditions. It is seen that gains of 5% to 15% are expected near the wind directions $255°$ and $265°$ at a turbulence intensity of 7.5%. Furthermore, smaller gains in the order of 5% can be expected for wind directions $220°$, $230°$, $240°$, $285°$ and $295°$ at a turbulence intensity of 7.5%. The look-up tables for higher turbulence intensities are included in Appendix A and indicate a strong decrease in expected gains for higher turbulence intensities.

FLORIS makes compromising assumptions about the wind farm terrain and wake behavior. Thus, these predictions hold a high uncertainty. As a first step to check its robustness, the optimized yaw angles from FLORIS are simulated in FarmFlow, the

in-house wind farm model of TNO (Kanev et al., 2018). FarmFlow is of the same fidelity of FLORIS, but has a different set of underlying equations and therefore provides different predictions. While FarmFlow predicts lower gains, which empirically is a common trend for FarmFlow compared to FLORIS, it also predicts little to no losses compared to baseline operation for most table entries, thereby solidifying confidence in the synthesized table of setpoints. Furthermore, after implementation in the real wind farm, the presented control module is toggled on/off every 35 minutes. This number is chosen such that toggling is not equal every day, thereby reducing dependency on diurnal variations in the atmosphere. Additionally, a lower toggling time would lead to less usable data due to postprocessing (step 5 of Section 4), and a higher toggling time would reduce number of measurements obtained under comparable atmospheric conditions. The optimal toggling time for such experiments remains uncertain in the literature.

## 4    Data processing

Sections 2 and 3 outlined the steps taken prior to the experiment. This section now addresses how the data is processed after the experiment. One-minute-averages of SCADA data are collected from August 19th, 2019 onward. Analysis was performed on data up until February 3rd, 2020. The data is postprocessed to eliminate any faulty or irrelevant entries as follows:

1. All data with SCADA-based wind direction estimates outside of the region of interest ($200°$ to $320°$) is discarded. Note that the plots in Section 5 will instead be cut off at a wind direction of $310°$ due to lack of data and yaw activity for higher wind directions.
2. All data with SCADA-based ambient wind speed estimates lower than 7 m/s and higher than 12 m/s is discarded, because of high noise levels and/or the optimized yaw angle setpoints being very small in these regions (Figure 6).
3. All data with SCADA-based turbulence intensity estimates lower than $12\%$ and higher than $18.0\%$ are discarded. The upper bound is because a high turbulence intensity reduces wake effects and thereby the expected gains. Moreover, a narrow turbulence intensity range is desired with as many datapoints as possible for a fair and statistically sound analysis, explaining the lower bound. The turbulence intensity range is on the higher side due to the nature of the experiment. The specified bounds allow for a sufficient number of measurements such that a sound statistical analysis can be performed.
4. All data where the turbines of interest produce less than 200 kW of power are discarded, to reduce the relative variance in power and eliminate any situations in which turbines exhibit cut-in and cut-out behaviour.
5. Data within 5 minutes after a toggle change (baseline vs. optimized operation) is discarded. Namely, due to the functioning of the turbine yaw controller, turbines do not instantly follow their yaw setpoint to limit usage of the yaw motor.
6. Power measurements are time filtered using a (non-causal) moving average with a centered time horizon of 5 minutes.
7. The datasets are separated according to their operational mode: baseline and optimized. The datasets are then balanced such that for each wind direction and wind speed (in steps of 1 m/s), the number of measurements for baseline operation and optimized operation are equal. This reduces bias in the analysis for unbalanced bins.

Note that a narrower wind speed and turbulence intensity range than used in this manuscript should, in theory, better quantify the change in power production due to wake steering. However, with the sparsity in the data set, further narrowing these ranges

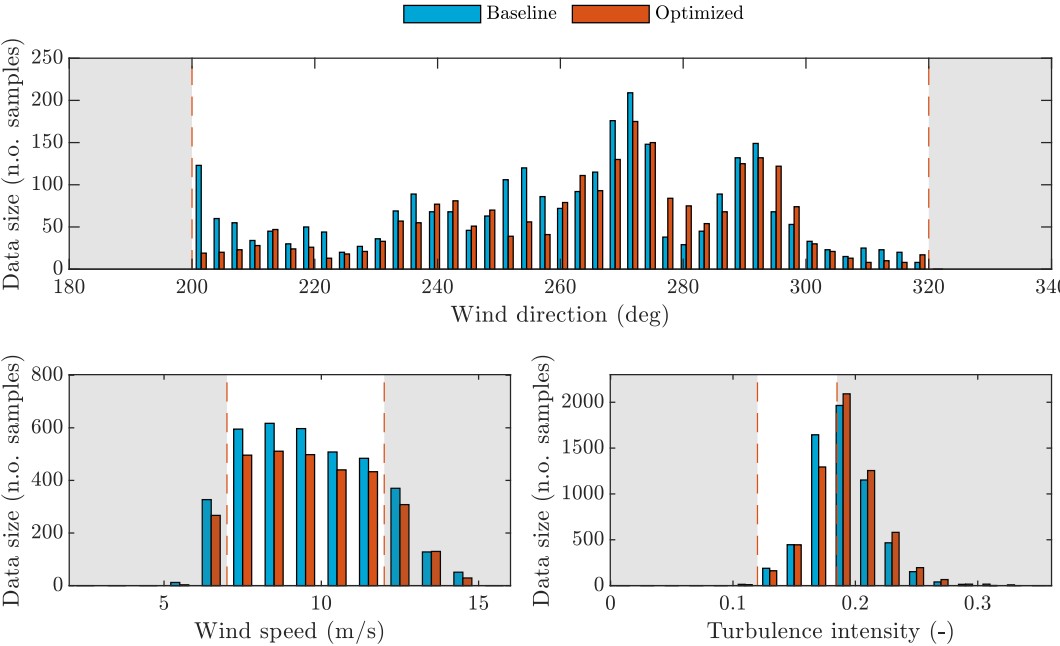

**Figure 7.** Filtered measurement data from 19 August 2019 until 3 February 2020, binned as a function of wind direction, wind speed, and turbulence intensity.

leads to a significant increase in statistical uncertainty. The current turbulence intensity and wind speed ranges are obtained
through an iterative process in pursuit of narrow uncertainty bounds and clear trends. With the filtered data, the energy ratio
method from Fleming et al. (2019) is then used to calculate the gains due to wake steering. Important to note is that WTG 10
and WTG 11 are curtailed to a maximum of 500 kW for long periods of time during the measurement campaign. To prevent the
elimination of this dataset, a part of the analysis is performed using the freestream-equivalent wind speed estimates of the local
wind turbine controllers, rather than the generated power signals. Note that the analysis for WTG 10 and WTG 11 is exclusively
done with measurements during curtailed operation, while the analysis for the other turbines relies on measurements during
normal operation – curtailed and non-curtailed measurements are not mixed within bins.

  Figure 7 shows the histograms of the postprocessed dataset, divided into *baseline* and *optimized* data. The relatively high
turbulence intensity shown in this figure corresponds to gains in power production in the order of 2% to 6% according to
FLORIS.

**5 Results & discussion**

This section analyzes the measurement data and quantifies the change in performance due to wake steering compared to
baseline operation. Note that all local wind speed estimates and power production signals are normalized with respect to the
measurements from WTG 25, to reduce the sensitivity of variables to the estimated ambient wind speed. Furthermore, 95%

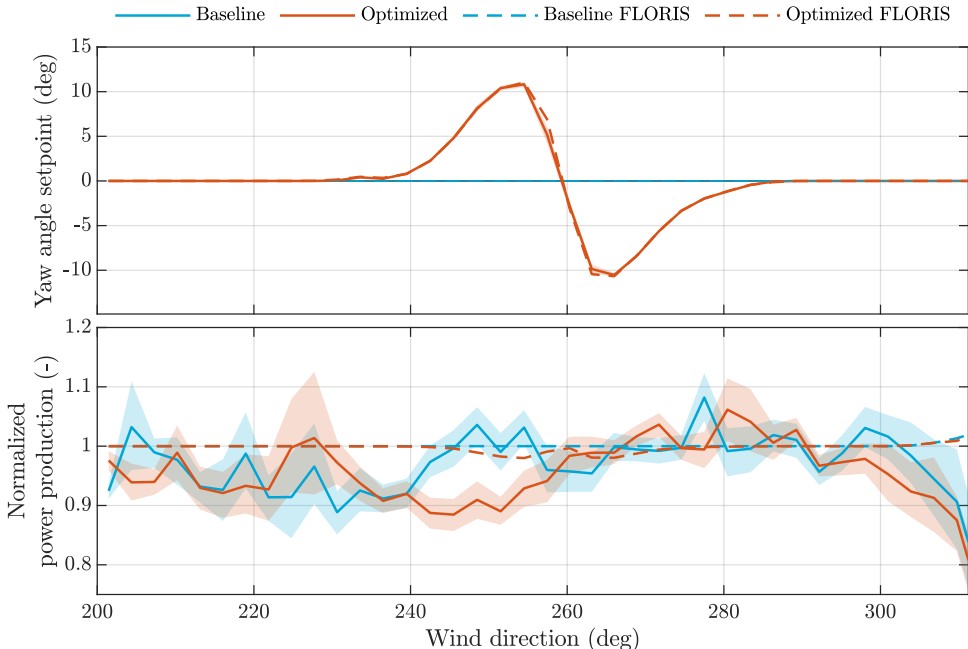

**Figure 8.** Yaw misalignments and corresponding power production for WTG 26, normalized with respect to WTG 25. The shaded areas show the 95% confidence bounds. The dashed lines represent the predictions for the measured inflow conditions by FLORIS. The number of samples in each bin is shown in Figure 7.

confidence intervals are calculated through bootstrapping (Efron and Tibshirani, 1993) for the results presented in this section.
Additionally, the results shown here are with respect to the estimated mean atmospheric conditions in front of WTG 26, derived from WTG 24 and WTG 25 as described in Section 3.

    Figure 8 portrays the yaw misalignment setpoints and the power production of WTG 26. The dashed lines represent the predictions from FLORIS, and the solid lines represent the measurements. Since WTG 26 is not misaligned for wind directions lower than 230 degrees and higher than 290 degrees, the normalized power production should equal to 1.0, as reflected in the
FLORIS predictions. Around wind directions of $255°$ and $265°$, yaw misaligments are assigned to the turbine, expected to lead to a loss in its power production. Looking at the measurements, the yaw setpoints are successfully assigned for all wind directions. However, the predicted loss in power production due to yaw misalignment is not reflected in the measurements. Rather, it appears that positive yaw misalignment angles lead to a significant decrease of about $10\%$ in the power production (wind directions of $240 - 250°$), while negative yaw misalignment angles even lead to a slight increase in the power produc-
tion compared to baseline operation (wind directions of $255 - 295°$). This indicates asymmetry and a high sensitivity in the power curve for yaw misalignment, which are both not accounted for in FLORIS. These observations were confirmed with measurement data from a different GE 1.5s turbine, briefly addressed in Appendix B. It may be that this asymmetry is partially due to bias in the wind vane sensor and consequently in the wind direction estimate. Literature suggests that a bias in these

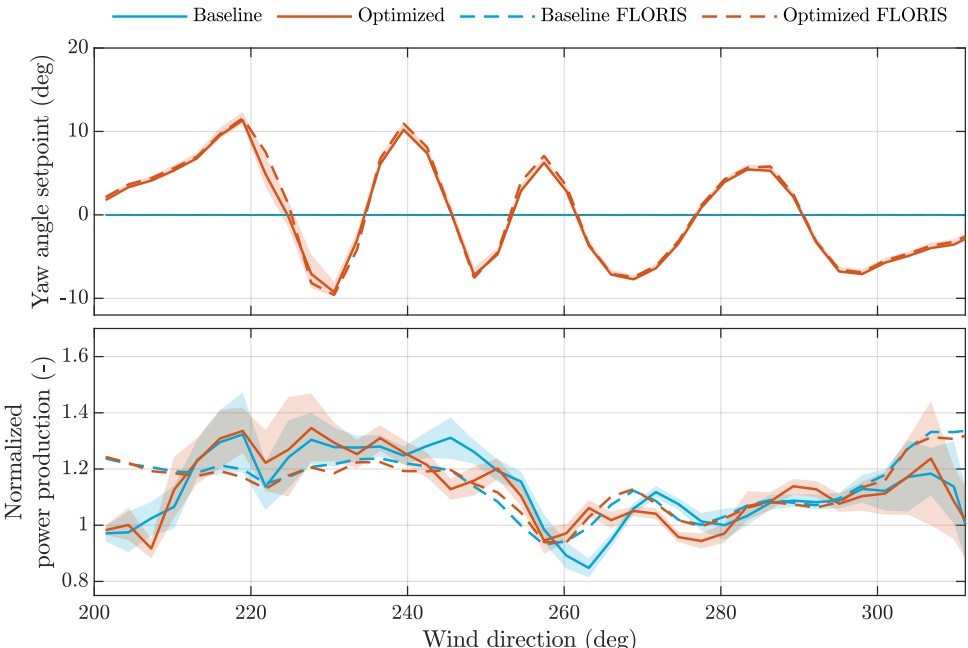

**Figure 9.** Yaw misalignments and power production for WTG E5, normalized with respect to WTG 25. The shaded areas show the 95% confidence bounds. The dashed lines represent the predictions for the measured inflow conditions by FLORIS. The number of samples in each bin is shown in Figure 7.

measurements is common in operational wind turbines (e.g., Fleming et al., 2014; Scholbrock et al., 2015; Kragh and Hansen, 2015), and the claim is further supported by the relatively large uncertainty seen in Figure 8. Furthermore, wind shear and veer are also known to skew the yaw-power curve (Howland et al., 2020), though both were quite benign in this experiment. More research is necessary to explain this yaw-power relationship for WTG 26. During the experiment, the wind direction bias was addressed by comparing what FLORIS predicts to be the wind direction where the largest wake losses are at downstream turbines to the actually measured power losses and wind directions. Though, it is not unreasonable to assume that this was insufficient. Moreover, Figure 8 shows that unknown factors lead to a systematically lower power production in the region $200 - 225°$ compared to WTG 25. Also, even though both datasets operate at zero yaw misalignment in the region $295 - 320°$, the *optimized* dataset shows a consistent loss compared to *baseline* operation for unidentified reasons. Hypothesized reasons for these discrepancies include terrain effects and differences in inflow conditions and turbine behavior between WTG 26 and WTG 25 to which the signals are normalized.

Figure 9 depicts the yaw misalignment setpoints and the power production of WTG E5. This turbine contains considerably more yaw variation between wind directions due to the close spacing and the scheduling of the considered downstream turbine (Table 2 and Figure 6). This figure shows that the yaw setpoints are applied successfully with little error. Further, note that the normalized power production for unwaked conditions is about $1.2$ instead of $1.0$ due to the larger rotor size and the higher

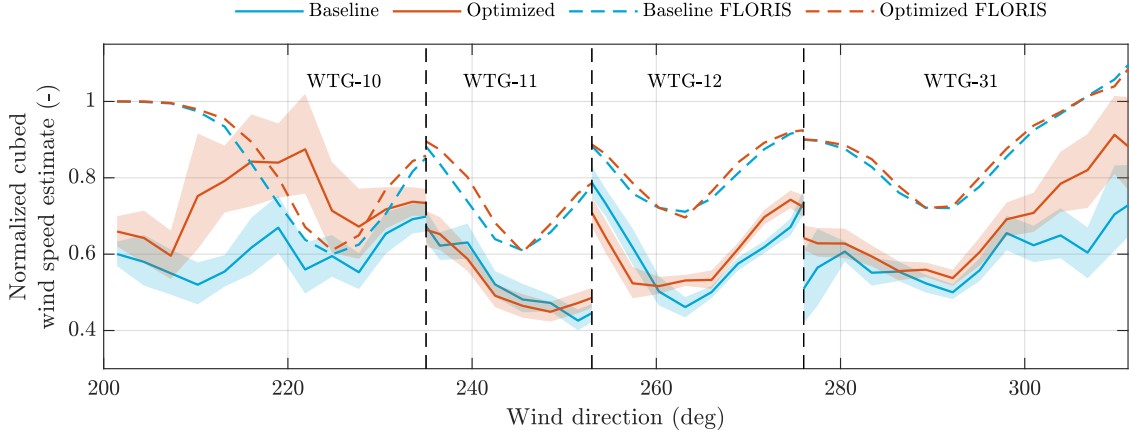

**Figure 10.** The cubed wind speed estimates of the downstream WTG of interest, serving as a surrogate for the power production under turbine derating. The results are normalized with respect to WTG 25. The shaded areas show the 95% confidence bounds. FLORIS underpredicts the wake losses. Moreover, the optimized dataset appears to outperform the baseline dataset, showing a benefit due to wake steering. The number of samples in each bin is shown in Figure 7.

tower of WTG E5. Note that the high relative power production at 310 degrees is due to WTG 25 operating in the wake of
WTG 24 and thereby producing less power, to which the power production of WTG E5 is normalized to. Moreover, wakes of
WTG 25 and WTG 26 cause losses in power production in both *baseline* and *optimized* operation for various wind directions
in Figure 9. These effects are both reflected in the measurements and seen in the FLORIS predictions. Notably, clear dips in
the power production for both *baseline* and *optimized* operation are seen at $260°$ and $278°$ caused by wake losses. FLORIS
predicts these losses, but lacks the accuracy to represent the finer trends in the measurements. Moreover, changes in the power
production due to a yaw misalignment on WTG E5 appear inconsistent (e.g., large loss at $245°$, no losses for $210°$ to $240°$)
compared to what was seen for WTG 26. The authors speculate that this may be either due to WTG E5 being of a different
turbine type than WTG 26, or due to different bias corrections in the wind vanes.

Figure 10 displays the cubed wind speed estimate of the downstream turbine of interest. The reason that this variable is
displayed instead of the power production is due to the fact that WTG 10 and WTG 11 are curtailed for long periods of time,
rendering the power measurements unusable. FLORIS predictions show a clear trend in power production losses due to wake
interactions of upstream turbines, notably at $225°$, $245°$, $265°$ and $290°$. Since none of the downstream turbines are yawed,
FLORIS predicts that optimized operation should never lead to any losses compared to baseline operation. When looking
at the measurements, this prediction is incorrect for WTG 11 and WTG 12. FLORIS is reasonably accurate in predicting at
what wind directions the largest dips in power production occur for downwind turbines. However, FLORIS overestimates the
wake recovery, and the power losses due to wake interactions are therefore larger than predicted. This suggests that FLORIS
predicts wake positions reasonably well, though lacks the accuracy to predict the subtle effects of a yaw misalignment. These
model discrepancies are hypothesized to be not in the least due to the lack of an accurate terrain model. Because of the

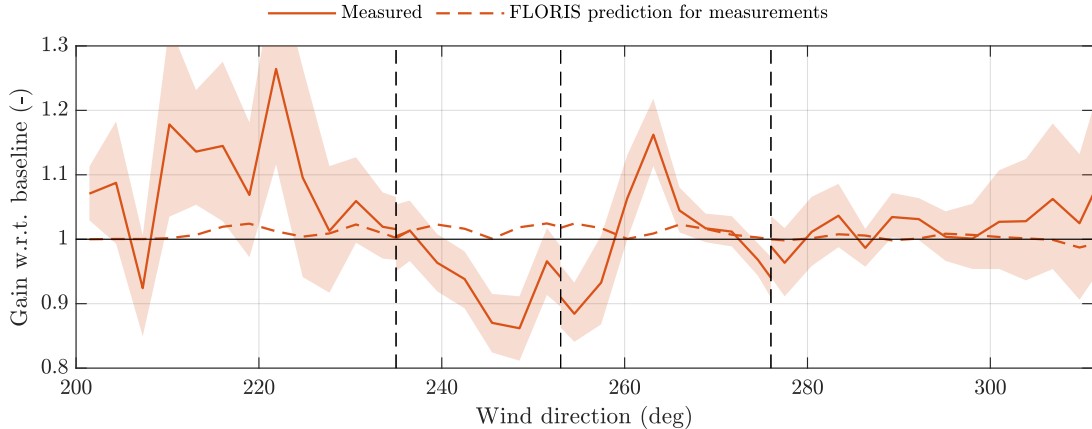

**Figure 11.** The estimated net gain of the three turbines for wake steering compared to baseline operation. The shaded area shows the 95% confidence bounds. The number of samples in each bin is shown in Figure 7.

underestimated wake effects in FLORIS, wake steering may have a higher potential than predicted, and the optimal yaw angles depicted in Section 3 may be underestimated. Moreover, the figure shows a very large increase in power production for the region $205 - 235°$ between optimized and baseline operation. This is due to WTG E5 steering away its wake from WTG 10. These two turbines are positioned closest together in the wind farm, and wake losses are therefore predicted to be the highest (Figure 1). Furthermore, gains in power production are seen in the region $260 - 320°$. This somewhat agrees with where FLORIS predicts gains to be. However, the measurements also show losses near $255°$. This is possibly due to the strong gradients in the yaw misalignment setpoints and thereby the sensitivity to noisy inflow conditions. Also, FLORIS predicts zero wake losses for a wind direction of $200°$ for both the baseline and optimized dataset, yet the measurements show a much lower wind speed. This is hypothesized to be due to topology effects and turbine interaction that was underestimated or not accounted for in FLORIS. The measurement uncertainty bounds are often larger than the potential gains predicted by FLORIS, which is largely due to poor the modeling performance of FLORIS rather than a high measurement uncertainty.

Finally, the change in performance for the combined three turbines is displayed in Figure 11. FLORIS predicts a relatively small but consistent gain across different wind directions of about 3%. This is largely due to high turbulence levels and the underestimated wake losses in FLORIS (Figure 10). This in turn leads to the underestimation of the benefits of wake steering. When looking at the measurements, a very large gain of up to 26% is seen at $222°$. Interesting to note is that this 26% gain is the situation where WTG E5 steers its wake away from WTG 10 (Figure 3), and WTG 26 has no influence on this interaction. If we only consider turbines WTG E5 and WTG 10, the combined gain in power production of turbines WTG E5 and WTG 10 is 35%. Though, it must be noted that the uncertainty bands are large for this bin. Generally, notable gains in power production are measured in the region $260 - 273°$ with a gain of 16% at $263°$, concerned with three-turbine-interaction. Interesting to note is that all three turbines experience an increase in power production for this wind direction, be it due to a yaw misalignment

or due to a steered wake. Among these three turbines, the largest gain comes from WTG E5 with a 29% increase in power by itself. Furthermore, Figure 11 also shows notable losses, especially in the region near 250°, due to large losses at WTG 26 originating from yaw misalignment and no gains downstream. Losses are also seen near the transition regions (black dashed vertical lines), possibly due to strong gradients in the yaw angles at these wind directions.

In addition to the mismatch between FLORIS and the actual yaw-power curve of WTG 26 and WTG E5, the lack of terrain effects in FLORIS are expected to have a significant impact on the results. This may be one of the key reasons for the overestimation of wake recovery in the FLORIS model, which in turn leads to an underestimation of the benefits of wake steering. Moreover, unmodeled effects such as secondary steering (Martínez-Tossas et al., 2018) may be a source of error. These unmodeled effects can have a positive effect on the success of wake steering. This leads to an underestimation of the potential benefits of wake steering and consequently to suboptimal yaw misalignment setpoints. Historical operational data may also be used to reduce the model-plant mismatch (Schreiber et al., 2019).

## 6 Conclusions and recommendations

This article presented a field experiment for wake steering at a commercial onshore wind farm in Italy. Three-turbine-interaction was considered, with the first two turbines operating under yaw misalignments to maximize the collective power production. The yaw setpoints were calculated according to an open-loop steady-state and model-based wind farm control solution. The field experiment shows significant gains, especially for two-turbine interaction, with an increase in combined power production of up to 35% for one particular two-turbine situation. Moreover, gains in power production for the three-turbine array up to 16% were measured for particular wind directions. However, the measurements also show notable losses for a region of wind directions, largely due to losses at the yaw-misaligned upstream turbines and due to insufficient or incorrect wake steering downstream.

Several important observations were made from the measurement data. Measurements shows that upstream turbines may benefit from nonzero yaw misalignment compared to the wind vane sensor, already leading to an effective increase in power production at these turbines without considering the phenomenon of wake steering downstream. Such effects have a large influence on the results presented in this article and are likely due to poor calibration of the wind vane sensors, rather than a physical property of the turbine. Moreover, the potential of wake steering was confirmed for a large range of conditions. The flatness of the turbine power curve effectively allows wake steering without losing much energy upstream. Also, while the surrogate model leveraged in this work is able to predict the dominant trends of wake interaction (i.e., FLORIS accurately predicts at what wind directions the wake losses are highest), large discrepancies are seen between its predictions and the field measurements. Notably, FLORIS assumes a symmetrical yaw-power curve of WTG 26 and WTG E5, assuming peak power production at zero yaw misalignment. In addition, FLORIS lacks important terrain effects and appears to overestimate wake recovery. Consequently, FLORIS underestimates the benefits of wake steering and the assigned yaw angles in this experiment are suboptimal.

At large, the following recommendations can be made for future wind farm validation trials:

– This article demonstrated the asymmetry and flatness one may find in the yaw-power curve of commercial wind turbines. This curve is particularly important to characterize accurately for wake steering. Therefore, future trials should perform experiments to allow such a characterization. At a higher level, this experiment showed the significant discrepancies between FLORIS and the measurements, especially at downstream turbines. One may want to perform simple and shorter wake steering tests to generate data for model tuning, such as keeping an upstream turbine and fixed yaw misalignment angles of $-20°$ to $+20°$ in steps of $5°$ for periods at a time under various wind shear and veer conditions (Howland et al., 2020). However, this may go at the cost of the plant's energy production and therefore also depends on the willingness of the wind farm operator. Doing such a model calibration may also indicate weaknesses in the model, such as the absence of ground effects, inaccuracies in the turbulence model, and variations in the surface level.

– A difficult trade-off must be made between the value of additional accurate measurements and the higher costs involved. Ideally, one would also measure the complete wake profiles downstream at a minimum sampling rate of 1 minute, measure the fluid density and atmospheric temperature at various heights from the ground, and identify the incoming turbulence levels. Furthermore, both for model tuning and the actual wake steering trials, an accurate characterization of the inflow conditions is essential, both in front of turbine WTG 26 but also in front of WTG E5. This could be achieved using lidar systems. Though, the authors cannot make a definitive conclusion on what equipment would provide most value and where it should be placed in a hypothetical future experiment. Rather, the scope of this article lies with the analysis of the experiment outcomes, rather than experiment design.

– Subsequently, an accurate baseline yaw controller that maximizes power production for the individual turbine is necessary to present a reliable baseline case to which the wake steering controller can be compared. Measured gains from wake steering should originate from gains at downstream rather than upstream turbines.

– Field campaigns should run for at least one year to minimize the impact of measurement uncertainty. Moreover, experiments ran throughout the year will provide a realistic idea of the efficacy of the tested concept and its impact on the annual energy production.

– In this experiment, which turbine was considered to be the "downstream turbine of interest" was decided according to the wind direction to maximize the potential benefits of wake steering. Unfortunately, this is expected to be the reason for poor performance near the transition regions. Such scheduling requires more research before implementation, and rather should be avoided whenever possible. Additionally, rather than smoothing the yaw angles with a Gaussian kernel to reduce yaw travel, it is valuable to look into solutions such as hysteresis (e.g., Kanev, 2020).

Finally, loads are neglected in this work, but play a vital role in adoption of the concept. Other noteworthy research topics to explore include dynamic models and the inclusion of heterogeneous inflow effects. In conclusion, this article supports the notion that further research is necessary, notably on the topic of wind farm modeling, before wake steering will lead to consistent energy gains in commercial wind farms.

*Code availability.* FLORIS is developed by CU Boulder, the Delft University of Technology and the National Renewable Energy Laboratory. A research-oriented MATLAB implementation is developed by Delft University of Technology, available at its Github repository (Doekemeijer and Storm, 2019). Note that a numerically efficient Python implementation of FLORIS is developed by the National Renewable Energy Laboratory, available at its Github repository (NREL, 2019). The work presented in this article uses the MATLAB implementation.

## Appendix A:  Additional look-up table figures

The turbine yaw setpoints were optimized for a large range of inflow conditions as described in Section 3.2. Figure 6 previously showed the optimal yaw setpoints for a low turbulence intensity of 7.5%. This appendix shows the optimal yaw setpoints for turbulence intensities of 13.5% and 18.0%.

The optimal turbine yaw setpoints for a turbulence intensity of 13.5% are shown in Figure A1. Compared to the situation with a turbulence intensity of 7.5%, the forecasted performance gains notably reduce. A higher ambient turbulence leads to more wake recovery, and thus the benefits of wake steering become less apparent. The optimal turbine yaw setpoints for a turbulence intensity of 18.0% are shown in Figure A2. Compared to the situations with turbulence intensities of 7.5% and 13.5%, the gains are very small. In practice, these gains are expected to drown in statistical uncertainty.

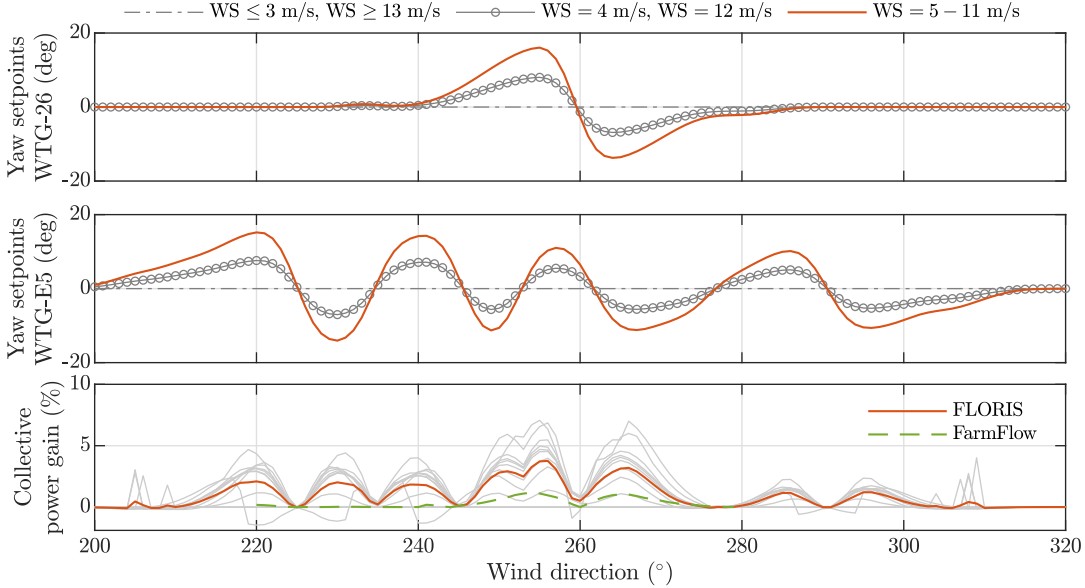

**Figure A1.** The optimal turbine yaw angle setpoints for WTG 26 and E5 for a freestream turbulence intensity of 13.5%. The averaged collective power gain of WTG 26, WTG E5 and the downstream machine (WTG 10, 11, 12, or 31) is shown as the solid orange line (FLORIS) and the green dashed line (FarmFlow) in the bottom plot. The gray lines therein represent the predicted gains for one wind speed by FLORIS.

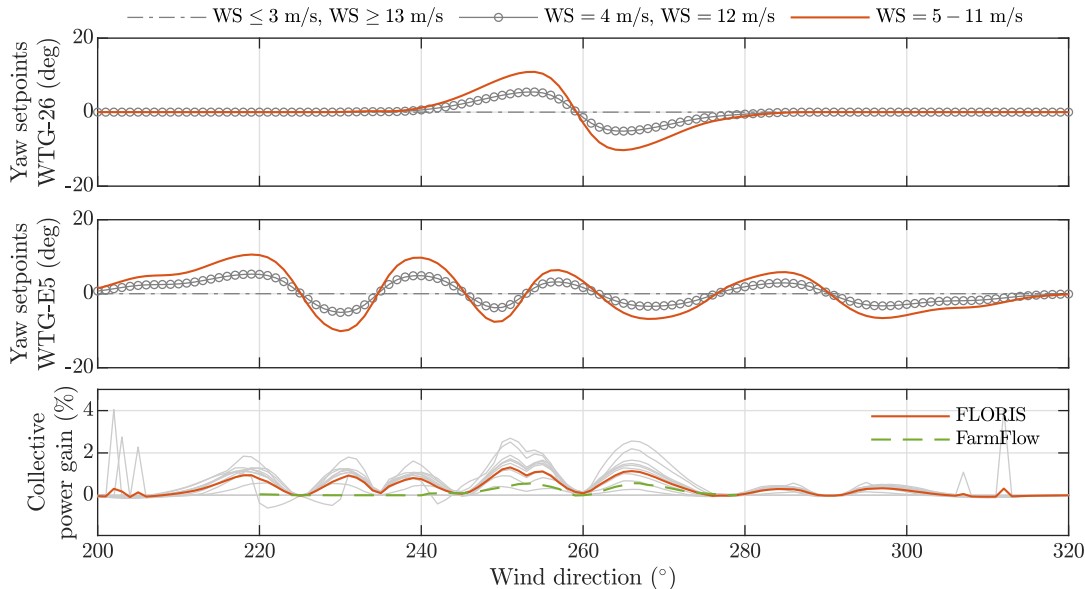

**Figure A2.** The optimal turbine yaw angle setpoints for WTG 26 and E5 for a freestream turbulence intensity of 18.0%. The averaged collective power gain of WTG 26, WTG E5 and the downstream machine (WTG 10, 11, 12, or 31) is shown as the solid orange line (FLORIS) and the green dashed line (FarmFlow) in the bottom plot. The gray lines therein represent the predicted gains for one wind speed by FLORIS.

## Appendix B: Yaw-power relationship for a GE 1.5s turbine

The experimental results from Section 5 indicate that negative yaw misalignment in WTG 26 leads to very small losses and sometimes even to a power gain compared to aligned operation. This behavior is verified by studying experimental data from a different GE 1.5s turbine inside the Sedini wind farm that is not included in the wake steering experiments: WTG 30.
SCADA data of this turbine is used to plot the normalized power production of the turbine against its yaw misalignment angle, shown in Figure B1. This figure shows that there is practically no decrease in power production when misaligning the turbine in the negative direction by less than $10°$. It is likely that this asymmetry is partially due to bias on the wind direction measurement, which has been seen more often in operational wind turbines as reported in the literature (e.g., Fleming et al., 2014; Scholbrock et al., 2015; Kragh and Hansen, 2015). Furthermore, wind shear and veer are also known to skew the yaw-
power curve Howland et al. (2020), though both were quite benign in this experiment. More research is necessary to explain this yaw-power relationship for WTG 26. During the experiment, the wind direction bias was addressed by comparing what FLORIS predicts to be the wind direction where the largest wake losses are at downstream turbines to the actually measured power losses and wind directions. Though, it is not unreasonable to assume that this was insufficient. The observations made for this wind turbine are in agreement with the behavior seen in WTG 26 and explains the large gains around the $260 - 280°$
region in the field experiments shown in Figure 11.

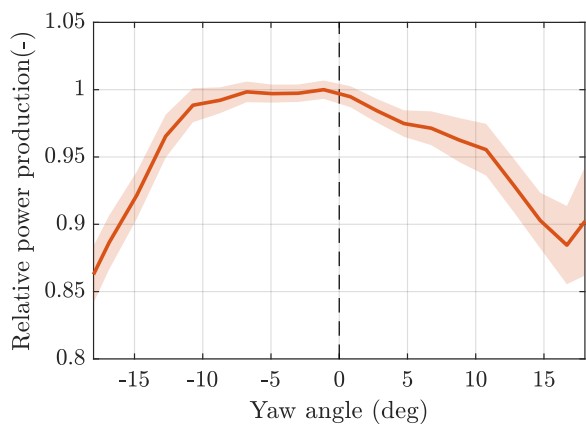

**Figure B1.** Relationship between the normalized power production and the yaw misalignment angle for an arbitrary GE 1.5s wind turbine in the Sedini wind farm. The data was collected for the range of 6 m/s to 12 m/s wind speeds. The asymmetry is clearly seen. Moreover, negative yaw misalignment shows a much smaller loss or even a very slight gain in power production compared to positive yaw misalignment.

*Competing interests.* The authors declare that they have had no competing interests in executing and publishing this work.

*Acknowledgements.* The authors thank Paul Fleming for providing guidance on processing the measurements retrieved from the field campaign. Moreover, the authors thank Marcus Zettl, Axel Busboom and Maria Gomez for their help in the early preparation phase of the experiment. Also, the authors thank Michael Howland for making the authors aware of the effect of wind shear and wind veer on the yaw-power curve of commercial turbines. Any mistakes in this work remain the authors' own. This work is part of the European CL-Windcon project, and has received funding from the European Union's Horizon 2020 research and innovation program under grant agreement No 727477.

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
