# Peer review of "Field experiment for open-loop yaw-based wake steering at a commercial onshore wind farm in Italy"

_Wind Energy Science, 2020_

## Referee Comment (RC1) · David Verelst (Referee) · 27 Jul 2020

Dear Authors,

Thank you for your contribution. I would like to acknowledge the challenges that always seem to arise when comparing measurements with simulations.

An interesting experimental dataset is compared with results from an equally inter-esting optimisation problem in which the authors present an open loop yaw steering control strategy with the aim to increase the net energy output of the wind farm. However, as a reader and reviewer, I am not convinced by all arguments with respect to the interpretation of the measurement and its comparison with the modelled results.

I've tried to summarize some of my concerns and questions below. Although some questions might be interpreted as an invitation to write another chapter or even other publication, that is not their intention. I appreciate the effort to keep the scope of the publication to a manageable level.

Elements that require brief additional considerations

- Based on figure 8, the measurements show a relatively large difference between the baseline and optimized case (up to a 10% increase in power production with wake steering), while the modelling shows only a 1-2% increase. This seems to indicate that for this turbine (WTG26) the wake steering is having a clear effect. However, in figure 9, for WTG E5, this can not be replicated. Why is that?

- You acknowledge loads are important in one short sentence all at the end. If they are important (of which I am quite convinced they are), shouldn't you also mention this either in the abstract and the literature review? I think it is perfectly acceptable to limit the scope of this publication to study the effects on power production. However, assuming that the wake steering concept can not be considered in a real production environment without taking loads into account, it should take a more visible role in the evaluation of your experiment and publication (in my opinion at least).

- Is there a specific reason why wake meandering is not discussed? I would expect wake meandering to be an important element in the context of wake steering, and it will have an impact on both power production and loads (considering partial wake conditions have a big impact on fatigue loads). How have you, or have others in the past, considered wake meandering when studying wake steering?

Since the modelling you present is based on steady state wake deflections, how would you expect (qualitatively) wake meandering to impact the power production when compared to a steady state modelling approach? Could this be an important focus area for future work?

- Have you considered any uncertainty (and/or a potential bias) in the wind direction measurement. If so, how would that affect the interpretation of figure B1 in particular, and the measurements in more general?

- If you where to re-plan the experiment again knowing what you know today, would you design it differently? Or in other words, based on your experience, how would you plan a follow-up experiment to address the challenges you have encountered?

- You briefly mention the measurements are in complex terrain. Could you elaborate further on why this might be very challenging in a validation measurement campaign?

Specific comments (see also marked comments in the annotated PDF):

- **Lines 49-50:** Sounds contradictory. I think it is quite clear why wake steering is not likely to affect the net energy production of wind farms in general. Howland et al (2019) for example summarize this quite well in their abstract/introduction. I understand that for certain cases (specific layout at a specific wind speed and wind direction) a dramatic power output can be obtained when employing wake steering and evaluating the effect on power production with a steady state wake deflection model. I also believe it is important to study that. However, I don't think it is correct to claim at this point in time that wake steering has a real potential to increase the net energy production of wind farms in general.

[Figure]

- **Lines 79-81:** A minor detail of course, but I think you can leave this statement out as it is not relevant for the paper. It also sounds like a snippet from the companies advertisement brochure ("global leader", "forefront").

- **Figure 2:** based on the [-] unit I assume data size is normalized in order to avoid disclosing too much sensitive information? Or is that referring to number of 10 minute averages (so unit is number of samples)?

- **Lines 102-104:** The surrogate model is based on a physical model I assume? One element is what the physical model can capture, the other how well the surrogate can re-capture the underlying data. To what is this statement referring to? This only becomes clear on line 155, maybe here you could refer to section 3.2 for more details on the surrogate model?

- **Line 112:** Completely agree, but as you already point out, a high quality reference measurement for wind speed, turbulence intensity and wind direction is required in a validation study context.

- **Lines 117-118:** Could you elaborate a little bit more what the context of these simulations are (model type, etc)?

- **Lines 139-142:** To my knowledge, the biggest hurdle would be certification, is that correct? I completely agree with the authors that a closed loop controller would be much more complex, however, technically it would not be prohibitively complex.

  I would imagine a wind turbine manufacturer will not open the controller up to its customers to perform these types of experiments. Knowing that the load certification process heavily relies on a well tuned controller, this is a reasonable precaution from the manufacturer side.

- **Figure 4:** Is each data point the 1 minute averaged wind speed?

[Figure]

Can you show a similar plot for the turbulence intensity and the wind direction?

Assuming you have access to the MET mast data, does it illustrate that, as you write on lines 120-125, it is simply too far away to be used reliably as an indication of free stream wind direction and speed?

Do you expect that this validation/correlation curve is independent of wind direction?

- **Lines 175-176:** I understand it is out of scope for this publication, but is it possible to add one or two sentences how different the "underlying equations" between the FLORIS and FarmFlow solvers are?

- **Lines 176-177:** For someone who is not a wind farm flow modeller expert this statement might not be obvious. Are there any references to back this statement? Or is it based on the author's general experiences as users and/or expert modellers?

- **lines 197-199:** Wouldn't this still result in a relatively broad range of operating an inflow conditions? I would be worried that all that averaging and aggregation will make it very difficult to make any clear conclusions since it won't be clear what exactly happens at which conditions.

How do the results compare for much more limited datasets? For example, for a given sector/wind speed bin/turbulence intensity bin for which you have a fair amount of measurement samples? That could illustrate in a more detailed manner how the wake steering is visible in the measurements.

- **Lines 200-205:** Based on figure 3, I can see that WTG 10 and 11 are downstream turbines. I am confused with what this means for your measurements? If the turbine is being curtailed, I would expect that it would be less affected by the wakes upstream, is that correct?

Are you at liberty to share to what exactly the curtailed operation refers to (in terms of different pitch and RPM strategy)?

- **Figure 8:** How many data samples do you have in each bin, I assume that varies per bin? Is the range of turbulence intensities and wind speeds similar across the bins?

  How does the yaw measurement uncertainty compares to the applied yaw error?

- **Figure 10:** The model predicts a very small difference (baseline vs optimized), while the measurements show a very different picture. Further, the difference between the models seems to be smaller than the 95% confidence of the measurements.

- **Figure 11:** could you also indicate the average net gain (simple average over the considered directions, not including probability of occurrence)? It seems to be positive, is that correct?

- **Lines 285-287:** I am puzzled by this statement and it contradicts the general understanding I have of wind turbines. Wind turbines operating under a yaw error will have by definition less power output and are subject to higher fatigue loads. So how can it be beneficial to operate a turbine with a constant yaw error?

  I can understand the statement when assuming there is a bias in the wind direction measurements, or that the complex terrain results in a flow field that is very complex and produces counter-intuitive results. However, I don't see such a discussion here.

- **Line 289:** You place "free" between quotes, but what about loads?

- **Lines 290-291:** If there are large discrepancies between the measurements and the model, how can you conclude the surrogate model is able to predict the dominant wake interaction trends? I understand that the model is intended to do so

(and effectively does in other cases), but that doesn't mean it is true for your specific case.

- **Line 313:** Agreed. I would suggest to split the conclusions chapter into 2: "conclusions" and "future work". In the "Future Work" section you could consider being more specific about what you suggest should be done to resolve specifically the shortcomings you've seen in your experiment. This could be valuable for future validation campaigns of the wake steering concept.

- **Figure 1B:** Is this a reasonable or comparable power-yaw curve when compared to other experiments, for example when looking at the data from Danaero, MEX-ICO, MexNext, etc?

  How have you verified the presented result is not due to a bias in the yaw inflow measurement?

Please also note the supplement to this comment:
https://wes.copernicus.org/preprints/wes-2020-80/wes-2020-80-RC1-supplement.pdf

---

## Referee Comment (RC2) · Anonymous Referee #2 · 3 Aug 2020

Dear Authors, your paper presents results from a wake steering experiment considering three-turbine interactions in complex terrain. It is well written and organized. The paper contains a unique wind farm control experiment and the work presented is very important for the wind energy research community. Thanks for working on it! It nicely confirms that there is a large potential for wind farm control, but also still more research is necessary to understand all effects.

In general, the paper could focus more on these effects which are not fully understood. For example in Section 5, line 221, the authors try to interpret the effect, although the uncertainty of the data is very high: you write: "...while negative yaw misalignment

angles even lead to a slight increase in the power production...". However, the difference between both curves for 255-295 deg is similar to the difference between both curves below 230 deg, where both lines should be equal, since WTG 26 is not misaligned. Therefore, the conclusion "that upstream turbines may benefit from nonzero yaw misalignment, already leading to an effective increase in power production at these turbines without considering the phenomenon of wake steering downstream" is hard to follow. Further, it might be that the upstream wind turbine already had a static yaw misalignment and a demanded nonzero yaw misalignment unintendedly aligned the turbine into the wind and thus increased the power. Another example is that for Figure 10, you write "the predictions (no losses due to wake steering for downwind turbines) are largely reflected", but for a quite a large are, there are losses for WTG 11 and 12. Focusing on these effects might help more to improve further wind testing campaigns compared to highlighting (sometimes uncertain) positive effects.

Further, there are several points where more details might help to better understand the work: - Section 3.1: You pointed out that the most important variable of the ambient condition is the wind direction. However, in Figure 4 you compare the wind speed from the lidar to the ones estimated by WTG 24 and 25. Further, Section 3.1 is relatively short. It would be interesting to know (if this information can be shared): * how and on which signals wind speed, wind direction and TI are estimated. * Further, it is not clear in Figure 4 if datapoints are 1 min or 10 min averages. * The lidar position could be included in Figure 3. Was it installed outside of the induction zone (based in standards more than 2.5 D) and was the data set filtered (e.g. sectors with wakes excluded)?

- Section 3.2: More details about the optimization would be helpful: you mentioned that the yaw setpoints have been optimized in steps of 1 m/s, but then they are fixed between 5 and 11 m/s. Are the values based on an average? And maybe you could also use TI=13.5% since in the experiment the lower bound is 12 %.

- Gaussian smoothing kernel: It definitely serves its purpose (reduces sensitivity) and looks fine in general. But at the "most important point", in a full wake situation (e.g. at

225 deg, WTG 26 in wake of WTG E5) in produces a setpoint of zero degree. Some comments of this drawback would be helpful, e.g. wouldn't a hysteresis or similar be more helpful?

- FarmFlow comparison: Results would be interesting, e.g. add line averaged over all wind speeds to Figure 6?

- Implementation: Here, more details than the last sentence in Section 3.2 would be helpful, e.g. * You describe, how the demanded yaw setpoint is derived from the estimated wind speed and wind direction via interpolation in a look-up-table. But it is not clear, if the estimated TI is used and if so, how? Sorry, if I missed it. * Why the controller is toggled every 35 min? * How is the demanded yaw setpoint added to the turbine? As a real setpoint or by having an offset to the measurement signal? How is the signal filtered? * If toggled off, is the turbine yawing instantaneously back or some time due to filtering? * How is the decision based on WTG 24 and 25 transferred to WTG E5 and 26? * Why there was a curtailment?

- Section 5: * Since you have been using the averaged estimates of WTG 24 and 25, wouldn't it be more consistent to use this average also for the postprocessing? * And the yaw angle setpoints are shown. Wouldn't be the yaw misalignment be more interesting, since usually wind turbines don't follow the setpoint instantaneous? Maybe it could help to understand the effect between 295-320 deg. * Figure 9: why is the baseline from Floris not 1.3 as stated in the text for unwaked conditions, e.g. 200-240 deg? * Figure 11: How does the Floris prediction here corresponds to the ones in Figure 6, A1, A2? If there has been a scheduling on TI, one would expect an average. However, close to 310 deg there is a prediction of losses, not present in Figure 6, A1, A2.

- Conclusions: It is not clear, why the "transition regions" lead to poor performance. Are those not only part of the postprocessing? Floris should optimize the yaw angles without this concept.

Minor comments: - l25f: Measurement uncertainties should be also present in wind tunnel experiments. Do you mean that the wind turbines are not measuring the wind direction by themselves? Would be good to specify.

- l87, Figure 2: Predominant wind directions seem to be west and south-east (and not south- west).

- l165, Figure 6: $\gamma$ has not been introduced. Maybe use yaw setpoint instead?

- l171: Maybe add 285+295 (WTG 31 in wake of WTG E5) to the list.

- Figures 8-11: you mentioned that the wind direction of interest are 200 to 320 deg. However, only 200 to 310 deg are shown.

- l256: It might be better to write "for the three-turbine-interaction", since the third turbine changes.

- l286. plural "s" missing for "these turbines".

---

## Author Comment (AC1) · 26 Oct 2020

Dear reviewers,

The authors express their gratitude to the reviewers for their time and efforts spent on providing accurate and constructive feedback to the submitted manuscript. Their comments play a crucial role in further improving the scientific quality and relevance of this work. In accordance to the provided feedback, the article has been revised. The objective of the attached document is to respond to all concerns raised and to outline the changes made to the manuscript.

[Figure]

Yours sincerely, Bart Doekemeijer

Please also note the supplement to this comment:
https://wes.copernicus.org/preprints/wes-2020-80/wes-2020-80-AC1-supplement.pdf

—————————————————

---

## Author Response (AR1)

| | |
|---|---|
| Date | October 26, 2020 |
| Our reference | WES-2020-80-RC1 |
| Contact person | B M Doekemeijer |
| E-mail | b.m.doekemeijer@tudelft.nl |
| Subject | Response to reviewers |

**Delft University of Technology**

Delft Center for Systems and Control

Address
Mekelweg 2 (3ME building)
2628 CD Delft
The Netherlands

www.dcsc.tudelft.nl

Reviewers
*Wind Energy Science Discussions, EWEA*

Dear reviewers,

The authors express their gratitude to the reviewers for their time and efforts spent on providing accurate and constructive feedback to the submitted manuscript. Their comments play a crucial role in further improving the scientific quality and relevance of this work. In accordance to the provided feedback, the article has been revised. The objective of the attached document is to respond to all concerns raised and to outline the changes made to the manuscript.

Yours sincerely,

Bart Doekemeijer

Enclosure(s):  Response to comments of Reviewer 1 (David Verelst)
Response to comments of Reviewer 2 (Anonymous)
Marked-up manuscript highlighting the changes made

**Response to Reviewer 1 (David Verelst)**

Dear reviewer. Thank you for your valuable comments. They play a role in improving the scientific relevance and clarity of this manuscript. We have addressed both your major and specific remarks in the remainder of this document.

**Major comments:**

Q1. *Based on figure 8, the measurements show a relatively large difference between the baseline and optimized case (up to a 10% increase in power production with wake steering), while the modelling shows only a 1-2% increase. This seems to indicate that for this turbine (WTG26) the wake steering is having a clear effect. However, in figure 9, for WTG E5, this can not be replicated. Why is that?*

A1. We appreciate that the reviewer raises this concern and he is correct in his statement that the effects of yaw misalignment on WTG 26 are not reflected in WTG E5 when comparing figures 8 and 9 of the original manuscript. The authors would like to point out that the power production shown in Figures 8 and 9 are solely the power production of the single, yawed turbine being WTG 26 (Figure 8) or WTG E5 (Figure 9), respectively. Based on Appendix B, it appears that the GE 1.5s turbines in this farm show a slight power increase for (measured) negative yaw misalignment, which also holds for WTG 26 as seen in Figure 8. These effects do not seem to reflect onto WTG E5, perhaps because this is a GE 1.5sle wind turbine and therefore not the same type as WTG 26. Another explanation might be that the turbines, coincidentally, have different bias correction terms. Additionally, it may be that shear and veer effects in the wind farm give rise to different yaw-power behavior [Howland et al., 2020]. Though, this is only speculation, and the authors cannot give a definitive answer to why the yaw-power curve differs in the way that it does between WTG 26 and WTG E5. To explain this, the authors have added a comment in Section 5: "The authors speculate ... bias corrections." and also made a remark on the additional effects that shear and veer may have on the yaw-power curve.

Q2. *You acknowledge loads are important in one short sentence all at the end. If they are important (of which I am quite convinced they are), shouldn't you also mention this either in the abstract and the literature review? I think it is perfectly acceptable to limit the scope of this publication to study the effects on power production. However, assuming that the wake steering concept can not be considered in a real production environment without taking loads into account, it should take a more visible role in the evaluation of your experiment and publication (in my opinion at least).*

A2. The authors appreciate the reviewer's comment and agree with him that this discussion should be more transparent in the article. Consequently, remarks and additional citations have been added to the abstract, introduction and conclusion.

Q3. *Is there a specific reason why wake meandering is not discussed? I would expect wake meandering to be an important element in the context of wake steering, and it will have an impact on both power production and loads (considering partial wake conditions have a big impact on fatigue loads). How have you, or have others in the past, considered wake meandering when studying wake steering? Since the modelling you present is based on steady state wake deflections, how would you expect (qualitatively) wake meandering to impact the power production when compared to a steady state modelling approach? Could this be an important focus area for future work?*

A3. The reviewer is correct in stating that wake meandering is an important factor in wake steering. The current steady-state wind farm models attempt to predict a 5-minute-average power production based on the mean inflow conditions and turbine yaw misalignment. Since FLORIS is tuned to high-fidelity data that includes wake meandering, FLORIS also captures the mean (time-averaged) effects of wake meandering on the power production. Naturally, the model is a simplification, but it does include wake meandering, among other effects. To include the full dynamic spectrum of wake meandering in the model, one would need to migrate towards the usage of dynamic wind farm models for control. This is an active field of research. Accordingly, an explanation of this has been added to the section on FLORIS and the recommendations have been updated.

Q4. *Have you considered any uncertainty (and/or a potential bias) in the wind direction measurement. If so, how would that affect the interpretation of figure B1 in particular, and the measurements in more general?*

A4. The reviewer points out an important factor in the controller design and data analysis. Currently, the wind direction is estimated by combining the local wind direction measurements from WTG 24 and WTG 25, considering WTG 26 is yawed and its wind direction measurement is less reliable. Note that the estimates from WTG 24 and WTG 25 are bias corrected by the internal GE algorithm. However, the workings and reliability of this bias correction algorithm are confidential, and the correctness of the assumption of using estimates of WTG 24 and WTG 25 for WTG 26 is uncertain. Indeed, Figure B1 suggests that WTG 26 may contain a bias in its wind direction measurement, and in that situation operates at a constant yaw misalignment when $\gamma = 0°$ is assigned. This would explain the power increase we see in Figure 8 when yawing the turbine in a negative direction. This was not modeled inside FLORIS and if it was, the optimal yaw misalignment profile would look quite different, most probably shifting emphasis to generally smaller and negative yaw misalignment angles. In the field campaign, an incorrect wind direction estimate also leads to the usage of the wrong yaw angle database entries, which may be a source of performance losses. Though, in defense of the authors, the wind directions at which the largest measured power deficit occurred at the downstream turbines (due to the wakes of WTG 26 and E5) were periodically compared to what FLORIS (plus bias corrections) predicts, making sure they align. With the data available, the authors believe this is the best they can currently do. A statement has been added to the manuscript indicating the likeliness of bias on the wind direction measurement and how that partially explains Figures 8 and B1.

Q5. *If you where to re-plan the experiment again knowing what you know today, would you design it differently? Or in other words, based on your experience, how would you plan a follow-up experiment to address the challenges you have encountered?*

A5. The authors attempted to answer this question by listing a number of "lessons learned" in the conclusion. The authors now understand that this does not exactly answer the same question. To clarify, one large source of error was the wind direction uncertainty and the large discrepancy between the measured and modeled yaw-power curve of turbine WTG 26 (and possibly also WTG E5). This curve must be characterized accurately in the model before performing the yaw optimization. Additionally, in general, it would be greatly beneficial to tune the simplified wind farm model to SCADA data before implementing a wind farm control solution. Specifically, figures 9 and 10 showed large discrepancies between FLORIS and the measurements. One may want to perform simple and shorter wake steering tests to generate data for model tuning, such as keeping WTG 26 and a constant yaw misalignments of $-20°$ to $+20°$ in steps of $5°$ for periods at a time and under various wind shear and veer conditions [Howland et al., 2020]. However, this may go at the cost of the plant's energy production and therefore also depends on the willingness of the wind farm operator. Doing such a model calibration may also indicate weaknesses in the model, such as the absence of ground effects and variations in the surface level. Important for model tuning is an accurate characterization of the inflow conditions, both in front of turbine WTG 26 but also in front of WTG E5. Note that a difficult trade-off must be made between the value of additional/more accurate measurements, and the additional costs involved. Ideally, one would also measure the complete wakes downstream at a higher sampling rate, measure the fluid density and atmospheric temperature at various heights from the ground, and identify the incoming turbulence levels. Though, the authors cannot make a definitive conclusion on what equipment would provide most value and where it should be placed in a hypothetical future experiment. Rather, the scope of this article lies with the analysis of the experiment outcomes, rather than experiment design. The authors believe, to appropriately address this question, a study would be necessary that would make a publication by itself. In this manuscript, we have extended and reformulated the conclusion to better resemble the answer presented here.

Q6. *You briefly mention the measurements are in complex terrain. Could you elaborate further on why this might be very challenging in a validation measurement campaign?*

A6. We thank the reviewer for attending us to the lack of clarity on this matter in the draft manuscript. Section 2.2 contains a list of challenges specific to this field experiment, among which the terrain complexity is mentioned. Specifically, this wind farm is situated in hilly area, where the turbines are positioned between 400 and 450 m above sea level. Such variations are likely to give variations in the ambient wind speed and wind direction between different upstream wind turbines. However, FLORIS assumes a uniform (homogeneous) ambient inflow, where each upstream turbine experiences the same wind speed, wind direction and turbulence intensity. Specifically, variations in the ambient wind direction have a large influence on wakes, and thereby on the wake steering campaign. Inclusion of such topology effects are an important challenge to tackle in future work. Additionally, as can be seen in Figure 1, various types of vegetation are present on the ground. The surface roughness varies with the type of vegetation, which in turn impacts the level of turbulence and thereby wake recovery. Such effects are not included in FLORIS and are speculated to play a role in the mismatch between the measurements and FLORIS for the downstream wind turbines. For clarification, this explanation has been summarized and included in Section 2.2.

**Specific remarks:**

SR1. *Lines 49-50: Sounds contradictory. I think it is quite clear why wake steering is not likely to affect the net energy production of wind farms in general. Howland et al (2019) for example summarize this quite well in their abstract/introduction. I understand that for certain cases (specific layout at a specific wind speed and wind direction) a dramatic power output can be obtained when employing wake steering and evaluating the effect on power production with a steady state wake deflection model. I also believe it is important to study that. However, I don't think it is correct to claim at this point in time that wake steering has a real potential to increase the net energy production of wind farms in general.*

A1. The authors appreciate the reviewer's comment, especially in relationship to the results of Howland et al. (2019). In the authors' eyes, wind farm control does still have real potential to increase the power production of wind farms on an annual basis. The authors believe that the measured increases for specific wake-loss-heavy situations may currently be outweighed by losses for other situations due to incorrect yaw misalignment. Namely, all persistent power losses can in theory be avoided by simply only yawing the turbines when an increase in power production can be guaranteed. The results in this publication support the notion that the current wind farm controllers do not suffice yet. An interesting observation supporting our opinion is that Siemens-Gamesa recently released their wind farm control solution "WakeAdapt", in which wake steering is sold as a service to wind farm operators to increase their annual power production. However, as the reviewer rightfully points out, this is not something we can guarantee. Therefore, we have softened the statement on lines 49-50.

SR2. *Lines 79-81: A minor detail of course, but I think you can leave this statement out as it is not relevant for the paper. It also sounds like a snippet from the companies advertisement brochure ("global leader", "forefront").*

A2. This statement has been removed from the manuscript.

SR3. *Figure 2: based on the [-] unit I assume data size is normalized in order to avoid disclosing too much sensitive information? Or is that referring to number of 10 minute averages (so unit is number of samples)?*

A3. We thank the reviewer for his detailed remarks, and have updated Figure 4 and Figure 7 accordingly. Indeed, the unit is the number of samples.

SR4. *Lines 102-104: The surrogate model is based on a physical model I assume? One element is what the physical model can capture, the other how well the surrogate can re-capture the underlying data. To what is this statement referring to? This only becomes clear on line 155, maybe here you could refer to section 3.2 for more details on the surrogate model?*

A4. We appreciate the suggestion made by the reviewer and have added the reference to Section 3.2 accordingly.

SR5. *Line 112: Completely agree, but as you already point out, a high quality reference measurement for wind speed, turbulence intensity and wind direction is required in a validation study context.*

A5. We thank the reviewer for his comment. We have addressed this issue previously with answering the major remark Q5 and the modifications made to the manuscript based on that comment.

SR6. *Lines 117-118: Could you elaborate a little bit more what the context of these simulations are (model type, etc)?*

A6. The authors assume that the reviewer is talking about the simulations shown in Figure 3. To clarity, the simulations are done with the same model parameters as used to generate the LUT, which are taken from the Renewable Energy publication of Doekemeijer et al. (2020). The turbulence intensity is 5% and the wind speed is 8.0 m/s. We understand that this level of turbulence intensity is not particularly realistic for this site, but Figure 3 should serve to explain how the turbine scheduling works in the work at hand, rather than give an accurate representation of the wake length and depth. A remark is added to the caption of Figure 3 to explain this.

SR7. *Lines 139-142: To my knowledge, the biggest hurdle would be certification, is that correct? I completely agree with the authors that a closed loop controller would be much more complex, however, technically it would not be prohibitively complex.*
*I would imagine a wind turbine manufacturer will not open the controller up to its customers to perform these types of experiments. Knowing that the load certification process heavily relies on a well tuned controller, this is a reasonable precaution from the manufacturer side.*

A7. The authors agree with the reviewer in that the main challenge currently lies with certification. While the loads can be kept under control by setting limits on the assigned yaw angles, other factors such as controller stability further complicate things. Moreover, very few closed-loop wind farm control solutions have been developed and tested in realistic (time-varying) simulations, and thus confidence levels for such solutions are still low.

SR8. *Figure 4: Is each data point the 1 minute averaged wind speed? Can you show a similar plot for the turbulence intensity and the wind direction? Assuming you have access to the MET mast data, does it illustrate that, as you write on lines 120-125, it is simply too far away to be used reliably as an indication of free stream wind direction and speed? Do you expect that this validation/correlation curve is independent of wind direction?*

A8. These are indeed the 1-minute averaged data points for the wind speed. The lidar system also provides wind direction measurements, with which we can make a similar comparison as for the wind speed:

[Figure]

When we look at this figure, we can see an offset in which the lidar-estimated wind direction is consistently lower than the turbine-estimated wind direction. However, we cannot know whether one or both of the estimates is wrong, and by how much. Instead, we opted to correct the turbine-estimated wind direction by comparing the situations of largest wake loss, as also answered in our response to major remark Q4. To the authors, this seems the most sensible way to tune the model, since this is concerned with the relative position of the wakes rather than the absolute values of wind direction estimates.

Moreover, unfortunately, the dataset does not include lidar estimates of the turbulence intensity, nor does it include measurements from the measurement tower. However, again, if the turbine-based wind estimates would not align with the measurements from the measurement tower, this would not imply that the turbine-based estimates are faulty, nor would it allow us to decide which of the two measurements is more reliable. The comparison shown in the publication provides supporting evidence that the wind speed (and with the figure here: wind direction) is roughly correct, though the conclusions we can draw from the data are limited.

Finally, the wind speed correlation curve (Figure 4) is not expected to be independent of the wind direction. Namely, ground effects can give rise to consistent higher/lower wind speeds in front of a turbine. The lidar system measures the inflow at a distance upstream of the rotor, while the turbine-based estimates are derived from the flow at the rotor plane. Therefore, a persistent difference between the estimates may arise for particular wind directions. Though, it is uncertain to the authors how large this effect would be.

Based on the reviewer's comment and our explanation here, we have updated Figure 4 and expanded the explanation in Section 2.3 on how the turbine-based estimates are calibrated.

SR9. *Lines 175-176: I understand it is out of scope for this publication, but is it possible to add one or two sentences how different the "underlying equations" between the FLORIS and FarmFlow solvers are?*

A9. We appreciate the reviewer's comment. Though, the authors refer readers that are interested in the model differences to the corresponding literature, rather than repeating such information in this article.

SR10. *Lines 176-177: For someone who is not a wind farm flow modeller expert this statement might not be obvious. Are there any references to back this statement? Or is it based on the author's general experiences as users and/or expert modellers?*

A10. We appreciate the reviewer's comment in improving the clarity and contributions of this manuscript. The authors assume that the reviewer's comment is concerned with the statement that FarmFlow has a common trend of predicting lower gains than FLORIS. This is indeed based on experience of the modellers/users of each model, which are co-authors on this publication. We understand that this statement can cause confusion and have therefore added that this is empirically motivated.

SR11. *lines 197-199: Wouldn't this still result in a relatively broad range of operating an inflow conditions? I would be worried that all that averaging and aggregation will make it very difficult to make any clear conclusions since it won't be clear what exactly happens at which conditions. How do the results compare for much more limited datasets? For example, for a given sector/wind speed bin/turbulence intensity bin for which you have a fair amount of measurement samples? That could illustrate in a more detailed manner how the wake steering is visible in the measurements.*

A11. We understand the concern of the reviewer and it is a valid one. Essentially, by clubbing different turbulence intensity and wind speed measurements together in a single bin, how can one assure that the reported gains are accurate? The authors believe that value remains in the averaged values reported from the bins. Due to the limited number of measurements, it is difficult to limit the bins to narrow ranges of turbulence intensity and wind speed. Rather, the turbulence intensity range has been limited to a range of $12\%$ to $18\%$, rather than from $0\%$ to $18\%$. Additionally, by normalizing the power production measurements to a reference turbine's power measurement (WTG 25), we largely remove dependency of power measurements on the freestream wind speed. Then, narrowing the number of samples would give more reliable results (reducing spread), but the limited number of samples may instead led to an increase in spread/uncertainty. Therefore, the authors iteratively found a relatively narrow range which produced narrow confidence bounds in the final plot. Individual values for very specific bins and ranges can provide more positive results than what was shown here, but would not be necessarily more accurate or representative of what can be gained with wake steering. The authors believe that the approach described in the manuscript shows a more realistic and representative picture of the potential of wake steering. In response to the reviewer's comment, a note has been added to Section 4 motivating the choice for the wind speed and turbulence intensity ranges.

SR12. *Lines 200-205: Based on figure 3, I can see that WTG 10 and 11 are downstream turbines. I am confused with what this means for your measurements? If the turbine is being curtailed, I would expect that it would be less affected by the wakes upstream, is that correct? Are you at liberty to share to what exactly the curtailed operation refers to (in terms of different pitch and RPM strategy)?*

A12. The reviewer is correct in stating that a downstream turbine will be less affected by an upstream turbine when curtailed *in terms of the power production*. Effectively, the freestream-equivalent wind speed at the downstream turbine is unchanged. Therefore, using estimates of the freestream-equivalent wind speed of the downstream curtailed turbines provides a very comparable measure to using the power production in noncurtailed operation. The word "freestream-equivalent" has been added in Section 4 to further clarify this. Also, unfortunately the authors cannot share how the turbines are curtailed for confidentiality reasons.

SR13. *Figure 8: How many data samples do you have in each bin, I assume that varies per bin? Is the range of turbulence intensities and wind speeds similar across the bins? How does the yaw measurement uncertainty compares to the applied yaw error?*

A13. The reviewer is correct in stating that the number of samples vary per bin, and that this does have a large effect on the statistical uncertainty of the reported outcomes in the manscript. To prevent repetition of plots, the authors decided to show the number of samples previously in Figure 7. Initially, the bins varying across wind directions do not contain an even distribution over turbulence intensities and wind speeds, as the reviewer rightfully points out. This was addressed by balancing the bins as stated in bulletpoint 7 of Section 4. For clarify, a reference to Figure 7 has been added to Figures 8-11.

SR14. *Figure 10: The model predicts a very small difference (baseline vs optimized), while the measurements show a very different picture. Further, the difference between the models seems to be smaller than the 95% confidence of the measurements.*

A14. The reviewer is correct in his observations. The authors agree that the surrogate model is not particularly accurate in predicting the wake losses at WTG-10, 11, 12 and 31. This is discussed in the text, for example, "However, FLORIS overestimates ... accurate terrain model." and "Also, FLORIS predicts ... accounted for in FLORIS." The fact that the confidence bounds seem larger than the potential gains predicted by FLORIS are more to blame on FLORIS rather than on the measurements. We have emphasized this in the text corresponding to Figure 10.

SR15. *Figure 11: could you also indicate the average net gain (simple average over the considered directions, not including probability of occurrence)? It seems to be positive, is that correct?*

A15. The average net gain when weighing each wind direction bin equally would be **1.7%**, which indeed is positive. However, the authors would like to refrain from such statements in the article, as it might be misleading and is not representative of the potential gain due to wake steering in this wind farm experiment.

SR16.    *Lines 285-287: I am puzzled by this statement and it contradicts the general under-standing I have of wind turbines. Wind turbines operating under a yaw error will have by definition less power output and are subject to higher fatigue loads. So how can it be beneficial to operate a turbine with a constant yaw error? I can understand the statement when assuming there is a bias in the wind direction measurements, or that the complex terrain results in a flow field that is very complex and produces counter-intuitive results. However, I don't see such a discussion here.*

A16.    We appreciate the reviewer's comment and agree with his statement that nonzero yaw misalignment should generally lead to power losses instead of power gains. This is what the authors also intended with this message, but in retrospect was poorly formulated. This phenomenon is likely due to a poorly calibrated wind vane sensor, rather than a true property of the wind turbine. We have added clarifications in the results section for WTG 26 and in the conclusion.

SR17.    *Line 289: You place "free" between quotes, but what about loads?*

A17.    We have rephrased this statement to better represent the discussions on the yaw-power curve relationship and on the loads previously discussed in major remark Q2.

SR18.    *Lines 290-291: If there are large discrepancies between the measurements and the model, how can you conclude the surrogate model is able to predict the dominant wake inter-action trends? I understand that the model is intended to do so (and effectively does in other cases), but that doesn't mean it is true for your specific case.*

A18.    We appreciate the reviewer's comment and agree that the results make it seem like the reliability of the entire FLORIS model is questionable. What we attempt to convey with our statement is that FLORIS is accurate in predicting at what wind directions the wake losses will be largest. Considering in this aspect the FLORIS model is accurate, then one could reason that FLORIS more or less accurately predicts where the mean position of the wake is as a function of wind direction. The actual depth of the wake is not estimated accurately, but perhaps this is secondary. If the position of the wake would be estimated incorrectly, that could lead to situations in which turbines are erroneously yawed and thereby perhaps accidentally steering a wake back onto a downstream turbine. If the error instead lies with the depth of the wake (as is the case in this manuscript), the result is a too large/small yaw angle, which typically has a much smaller effect on the success of the algorithm. We have attempted to clarify this in the manuscript, both in the Results section and the Conclusions.

SR19.    *Line 313: Agreed. I would suggest to split the conclusions chapter into 2: "conclusions" and "future work". In the "Future Work" section you could consider being more specific about what you suggest should be done to resolve specifically the shortcomings you've seen in your experiment. This could be valuable for future validation campaigns of the wake steering concept.*

A19. The authors appreciate the reviewer's comment and the corresponding major comment Q5. The conclusions have been extended to include a more detailed overview of recommendations for future experiments. Furthermore, the authors would like to refrain from introducing subsections in the conclusion. Namely, the conclusions section, as it is written now, would have to be split up into a *conclusions*, *recommendations*, and *wrap-up* subsection. The authors believe this would worsen the manuscript's readability.

SR20. *Figure 1B: Is this a reasonable or comparable power-yaw curve when compared to other experiments, for example when looking at the data from Danaero, MEXICO, MexNext, etc? How have you verified the presented result is not due to a bias in the yaw inflow measurement?*

A20. We thank the reviewer for his comment, and he is very right in pointing out that the presented bias may very well be due to a poor yaw inflow measurement rather than a physical property of the turbine. Additionally, wind shear and veer have been reported to effect the yaw-power curve of commercial wind turbines [Howland et al., 2020]. The authors have previously addressed this in their response to major remark Q4. In the revised manuscript, the authors explain that they cannot verify whether the power-yaw curve is due to this bias in the yaw inflow measurement with the data available, but that it is a likely assumption. Literature suggests a yaw bias is common in operational wind turbines [e.g., Fleming et al., 2014, Scholbrock et al., 2015, Kragh and Hansen, 2015]. The authors attempted to mitigate this bias by comparing at what wind direction the largest wake losses are at downstream turbines compared to the actually measured losses and wind directions. Though, it is not unreasonable to assume that this was insufficient. In addition to the adjustments made in response to Q4, remarks have been added in the results section and the appendix to further clarify this issue, including a reference to the literature mentioned in this response.

**Response to Reviewer 2 (Anonymous)**

Dear reviewer, thank you for your compliments, for reviewing the revised manuscript and for providing us with useful suggestions to improve this manuscript. We have split up your commentary in parts, attempting to address each of your concerns with care.

Q1. Dear Authors, your paper presents results from a wake steering experiment considering three-turbine interactions in complex terrain. It is well written and organized. The paper contains a unique wind farm control experiment and the work presented is very important for the wind energy research community. Thanks for working on it! It nicely confirms that there is a large potential for wind farm control, but also still more research is necessary to understand all effects. In general, the paper could focus more on these effects which are not fully understood. For example in Section 5, line 221, the authors try to interpret the effect, although the uncertainty of the data is very high: you write: ". . .while negative yaw misalignment angles even lead to a slight increase in the power production. . .". However, the difference between both curves for 255-295 deg is similar to the difference between both curves below 230 deg, where both lines should be equal, since WTG 26 is not misaligned. Therefore, the conclusion "that upstream turbines may benefit from nonzero yaw misalignment, already leading to an effective increase in power production at these turbines without considering the phenomenon of wake steering downstream" is hard to follow. Further, it might be that the upstream wind turbine already had a static yaw misalignment and a demanded nonzero yaw misalignment unintendedly aligned the turbine into the wind and thus increased the power.

A1. We thank the reviewer for his/her insightful comment. Namely, the reviewer addresses an important point that was also addressed by reviewer 1: a relatively large uncertainty remains in the power production for the baseline and optimized dataset shown in Figure 8 and, based on the general understanding of wind turbines, a nonzero yaw angle leading to a consistent increase in power production is counter intuitive and rather points towards an issue with the baseline wind direction estimate and yaw controller. The text in Section 5, Section 6 and Appendix B have been updated to further highlight the possibility is these phenomena being due to a bias in the wind direction estimates. Supporting literature for this claim has also been included in the manuscript.

Q2. Another example is that for Figure 10, you write "the predictions (no losses due to wake steering for downwind turbines) are largely reflected", but for a quite a large are, there are losses for WTG 11 and 12. Focusing on these effects might help more to improve further wind testing campaigns compared to highlighting (sometimes uncertain) positive effects."

A2. We appreciate the reviewer's comment and agree with him/her that he cited state-
ment is incorrect. What the authors intended to convey is that FLORIS does a
reasonable job in predicting where the largest losses will be (i.e., when we wake will
have the largest overlap with a downstream turbine). However, indeed, FLORIS
does not accurately predict losses at downstream turbines compared to baseline
operation. This statement has been adjusted and Section 5 of the manuscript has
been updated according to the explanation made in this response.

Q3. Further, there are several points where more details might help to better understand
the work:

Q3.1. Section 3.1: You pointed out that the most important variable of the ambient
condition is the wind direction. However, in Figure 4 you compare the wind
speed from the lidar to the ones estimated by WTG 24 and 25.

A3.1. The reviewer is very correct in his/her observation and we have adjusted
the manuscript accordingly. Namely, we have included a figure comparing the
wind direction estimate from the turbines with that from the lidar system. We
have also introduced a more elaborate discussion on the accuracy and valid-
ity of these estimates, and how this information is used for the field campaign.

Q3.2. Further, Section 3.1 is relatively short. It would be interesting to know (if
this information can be shared):

Q.3.2.1. how and on which signals wind speed, wind direction and TI are esti-
mated.

A.3.2.1. The estimates are derived by averaging the estimated quantities for WTG
24 and WTG 25, as reported in Section 3.1. Unfortunately, the authors
cannot share the functioning of these wind turbine estimators for con-
fidentiality reasons. Furthermore, a remark is added to Section 3.1 ex-
plaining how the wind direction estimates from the wind turbines were
monitored and corrected by comparing at what wind direction the largest
power dips occur at downstream turbines.

Q.3.2.2. Further, it is not clear in Figure 4 if datapoints are 1 min or 10 min
averages.

A.3.2.2. The reviewer raises an excellent concern. The datapoints from WTG 26
are 1-minute averages, while the datapoints from the lidar system are
10-minute averages. The authors agree that this is not clear to the read-
ers. Moreover, this also explains the notable spread in the plot. We have
included a comment in Figure 4 in the manuscript.

Q.3.2.3. The lidar position could be included in Figure 3. Was it installed outside of the induction zone (based in standards more than 2.5 D) and was the data set filtered (e.g. sectors with wakes excluded)?

A3.2.3. We appreciate the reviewer's comment and agree with him/her that more information about the lidar system in the field campaign is valuable. Therefore, the lidar system's position in included in Figure 1. This figure shows that the lidar system lies at a distance of about 2.5-3D in front of WTG 26, and therefore lies outside of the induction zone as suggested by the reviewer. Additionally, with the additional plot, it becomes clear that no special filtering has been applied to remove sectors with wakes (e.g., there are several measurements near a wind direction of $90°$). We have added a remark in the caption of Figure 4.

Q4. Section 3.2: More details about the optimization would be helpful: you mentioned that the yaw setpoints have been optimized in steps of 1 m/s, but then they are fixed between 5 and 11 m/s. Are the values based on an average? And maybe you could also use TI=13.5% since in the experiment the lower bound is 12%.

A4. We appreciate the reviewer's comment and agree that the current manuscript may cause confusion in how we ended up with the final lookup table. To clarify, optimal yaw angles were calculated for each wind speed, wind direction and turbulence intensity. Afterwards, these yaw angles were indeed averaged in the range of 5-11 m/s and smoothed. The final angles were verified by simulating them in FLORIS and reanalyzing the predicted power gains, which did not lead to noticeable losses compared to the initial angles. The authors have added clarifications in Section 3.2 to address this. Finally, the authors decided to proceed with to show the yaw angles and estimated power gains for the lower TI value in the main body of the manuscript to provide some theoretical estimated upper bound of the potential of the wake steering experiment.

Q5. Gaussian smoothing kernel: It definitely serves its purpose (reduces sensitivity) and looks fine in general. But at the "most important point", in a full wake situation (e.g. at 225 deg, WTG 26 in wake of WTG E5) in produces a setpoint of zero degree. Some comments of this drawback would be helpful, e.g. wouldn't a hysteresis or similar be more helpful?

A5.  The reviewer makes an excellent suggestion. Indeed, full wake overlap between WTG E5 and WTG 10 (we assume this is what we reviewer implied) occurs at a wind direction of 225 degrees. When sweeping over this turbine from a wind direction below 225 degrees to a wind direction above 225 degrees, the optimal yaw angle has a discontinuous jump near 225 degrees. At that point, it becomes more valuable to steer the wake to the opposite side of the wind turbine and therefore the optimal yaw angle for WTG E5 goes from a large positive number to a large negative number. This would cause large wear on the yaw actuators and therefore we smoothed the angles as described in the manuscript. Indeed, this effectively leads to a near-zero yaw setpoint at 225 degrees. As the reviewer rightfully points out, hysteresis would be a much better solution to this. However, the current framework provided by the turbine manufacturer did not allow for such an implementation. We have elaborated on this in Section 3 and additionally included it as a recommendation for future experiments.

Q6.  FarmFlow comparison: Results would be interesting, e.g. add line averaged over all wind speeds to Figure 6?

A6.  We appreciate the reviewer's comment and have included the predicted average power gain from FarmFlow in Figures 6, A1 and A2.

Q7.  Implementation: Here, more details than the last sentence in Section 3.2 would be helpful, e.g.

Q7.1.  You describe, how the demanded yaw setpoint is derived from the estimated wind speed and wind direction via interpolation in a look-up-table. But it is not clear, if the estimated TI is used and if so, how? Sorry, if I missed it.

A7.1.  We appreciate the reviewer's comment and agree that this may have been unclear in the manuscript. Actually, we interpolate the yaw angles over wind direction, wind speed and turbulence intensity. We have added a clarification in Section 3.

Q7.2.  Why the controller is toggled every 35 min?

A7.2.  We appreciate the reviewer's comment and understand that this is not clear to the reader. Actually, this number is chosen such that toggling is not equal every day, thereby reducing dependency on diurnal variations in the atmosphere. Additionally, a lower toggling time would lead to less usable data due to postprocessing, and a higher toggling time would reduce number of measurements obtained under comparable atmospheric conditions. The authors have added clarifications to the manuscript in Section 3.

Q7.3.   How is the demanded yaw setpoint added to the turbine? As a real setpoint or by having an offset to the measurement signal?

A7.3.   The yaw setpoint was assigned by adding an offset to the wind vane measurement, thereby "tricking" the turbine into yawing to a certain position.

Q7.4.   How is the signal filtered?

A7.4.   We assume this question relates to the estimated atmospheric quantities, being the wind direction, wind speed and turbulence intensity. Filtering and bias correction is part of the internal estimation algorithm of the wind turbines, and this information is not shared by the manufacturer. Additionally, averaging of the quantities between WTG 24 and WTG 25 provides some filtering.

Q7.5.   If toggled off, is the turbine yawing instantaneously back or some time due to filtering?

A7.5.   The turbine may take some time to yaw to their assigned setpoint due to the functioning of the yaw controller of the turbine. This is why, in postprocessing, data within 5 minutes after a toggle change was discarded. We have added a clarification in Section 4 on data processing.

Q7.6.   How is the decision based on WTG 24 and 25 transferred to WTG E5 and 26?

A7.6.   The reviewer clearly has eye for detail and the authors have indeed not explained this sufficiently in the manuscript. Actually, the estimated wind direction, wind speed and turbulence intensity of WTG 24 and WTG 25 are used to generate one mean ambient inflow wind direction, wind speed and turbulence intensity in front of WTG 26. These estimated quantities are used for interpolation to obtain setpoints for WTG E5 and 26. Note that the wind direction at WTG 26 is assumed to be equal to the wind direction of turbine E5. A part of the introduction of Section 3 has been rewritten for improved clarity.

Q7.7.   Why there was a curtailment?

A7.7.   Unfortunately this was without our knowledge and outside of our control. In an ideal situation, this curtailment would not have happened during our experiments.

Q8.  Section 5:

Q8.1.   Since you have been using the averaged estimates of WTG 24 and 25, wouldn't it be more consistent to use this average also for the postprocessing?

A8.1. The reviewer is exactly right and this is actually what the authors have done. We have added a clarifying statement to the first paragraph of Section 5.

Q8.2. And the yaw angle setpoints are shown. Wouldn't be the yaw misalignment be more interesting, since usually wind turbines don't follow the setpoint instantaneous? Maybe it could help to understand the effect between 295-320 deg.

A8.2. We completely agree with the reviewer that it would be more insightful to look at the achieved yaw angles rather than only the yaw angle setpoints. However, unfortunately, accurate yaw sensors for WTG 26 and E5 were only available during the first two months of the campaign. If we would produce a figure showing the estimated wind direction and nacelle yaw with the narrow bands on wind speed and turbulence intensity (following the regular postprocessing procedure as described in the manuscript), any similarity in the measured and predicted yaw curve is lost. Moreover, no useful plots can be made with the regular turbine wind vanes of WTG 26 and E5.
Using more data and different postprocessing may provide a curve that better resembles the assigned yaw curve, yet adding this to the manuscript would require an additional explanation of how the data is postprocessed, how that is different from the other data in the plot, and why. Furthermore, the bins would be based on different (less) data which further confuses the reader. To prevent further confusion in the manuscript, the authors decide not to include the additional lines in Figures 8 and 9.

Q8.3. Figure 9: why is the baseline from Floris not 1.3 as stated in the text for unwaked conditions, e.g. 200-240 deg?

A8.3. We appreciate the reviewer's excellent eye for detail. Actually, the baseline value for FLORIS in Figure 9 should be 1.2, after revising this simulation set-up in FLORIS. The authors initially believed it to be 1.3 because that was the highest value in the plot at a wind direction of approximately 310 degrees. However, at a wind direction of 310 degrees, the wake of WTG 24 starts impacting the power production of WTG 25. Thus, effectively the power production of WTG 25 is less. Since the power signal of WTG E5 is normalized to WTG 25, this wrongly raised the idea that the power production of WTG E5 was higher at this wind direction. We have corrected this in the manuscript and added a comment explaining the apparent increase in power production in Figure 9 at a 310 degrees.

Q8.4. Figure 11: How does the Floris prediction here corresponds to the ones in Figure 6, A1, A2? If there has been a scheduling on TI, one would expect an average. However, close to 310 deg there is a prediction of losses, not present in Figure 6, A1, A2

A8.4. The reviewer makes a rightful observation in that there are very slight losses predicted even by FLORIS at high wind directions. To clarify, the FLORIS predictions have been obtained by simulating the prescribed yaw setpoints and ambient conditions for each measurement in each bin, and then post-processing the outcomes in the same manner as with the measurement data. This process has also helped in finding errors in the postprocessing code. Notice that the predicted gains in Figure 11 share much similarity with the predicted gains in Figure A2. The loss at 310 degrees is likely to be explained with Figure 7. Figure 7 shows that there are very few datapoints in the bins above 300 degrees. Even though data entries are balanced within each bin to minimize dependencies on the wind speed and turbulence intensity, it may still happen that notable differences in the turbulence intensity or wind speed occur with sparsely populated bins. A narrower wind speed and turbulence intensity range may have been beneficial here, though it would probably lead to insufficient data to draw any conclusions. A discussion has been added to Section 4, which also answers Specific Remark 11 of Reviewer 1.

Q9. Conclusions: It is not clear, why the "transition regions" lead to poor performance. Are those not only part of the postprocessing? Floris should optimize the yaw angles without this concept.

A9. The appreciate the reviewer's comment. Actually, the transition regions are not only for postprocessing. Namely, if all turbines were included in the yaw optimization, then turbines (especially E5) would attempt to steer their wake in between downstream turbines. This is difficult and would lead to smaller yaw misalignment angles and smaller power gains than when following the approach used currently in the manuscript. This was previously addressed in Section 2.2. Consequently, since turbine scheduling was included in the yaw optimization, there are sudden transition points at which different turbines become the turbines of interest. This may cause strong gradients in the optimal yaw angle and thereby, after smoothing, are more sensitive to power losses. For clarification, a remark has been added to Section 3 stating that the optimization was done using the wind-direction-scheduled layout.

**Specific remarks**

SR1. l25f: Measurement uncertainties should be also present in wind tunnel experiments. Do you mean that the wind turbines are not measuring the wind direction by themselves? Would be good to specify.

A1. We agree with the reviewer and we have made modifications to the manuscript accordingly.

SR2. l87, Figure 2: Predominant wind directions seem to be west and south-east (and not south-west)

A2. We agree with the reviewer and we have made modifications to the manuscript accordingly.

SR3. l165, Figure 6: $\gamma$ has not been introduced. Maybe use yaw setpoint instead?

A3. We thank the reviewer for his eye for detail and have explained the symbol in the text.

SR4. l171: Maybe add 285+295 (WTG 31 in wake of WTG E5) to the list.

A4. We agree with the reviewer and we have made modifications to the manuscript accordingly.

SR5. Figures 8-11: you mentioned that the wind direction of interest are 200 to 320 deg. However, only 200 to 310 deg are shown.

A5. The reviewer is correct. Unfortunately, very few measurements are available at high wind directions. Additionally, there is no yaw misalignment at the wind direction range of 310-320 degrees. Therefore, we decided not to show these values in the final results. We have added a remark to Section 4 noting this.

SR6. l256: It might be better to write "for the three-turbine-interaction", since the third turbine changes.

A6. We appreciate the reviewer's remark and have made modifications to the manuscript accordingly.

SR7. l286. plural "s" missing for "these turbines".

A7. We thank the reviewer and have made modifications to the manuscript accordingly.

[revised manuscript text omitted]